# The TLR7/9 adaptors TASL and TASL2 mediate IRF5-dependent antiviral responses and autoimmunity in mouse

Ales Drobek ®[1], Léa Bernaleau ®[1], Maeva Delacrétaz ®[1], Sandra Calderon Copete[2], Claire Royer-Chardon[3], Mélissa Longepierre[1], Marta Monguió-Tortajada ®[1], Jakub Korzeniowski ®[1], Samuel Rotman ®[3], Julien Marquis ®[2] & Manuele Rebsamen ®[1] ✉

Endosomal nucleic acid sensing by Toll-like receptors (TLRs) is central to antimicrobial immunity and several autoimmune conditions such as systemic lupus erythematosus (SLE). The innate immune adaptor TASL mediates, via the interaction with SLC15A4, the activation of IRF5 downstream of human TLR7, TLR8 and TLR9, but the pathophysiological functions of this axis remain unexplored. Here we show that SLC15A4 deficiency results in a selective block of TLR7/9-induced IRF5 activation, while loss of TASL leads to a strong but incomplete impairment, which depends on the cell type and TLR engaged. This residual IRF5 activity is ascribed to a previously uncharacterized paralogue, *Gm6377*, named here TASL2. Double knockout of TASL and TASL2 (TASL[DKO]) phenocopies SLC15A4-deficient *feeble* mice showing comparable impairment of innate and humoral responses. Consequently, TASL[DKO] mice fail to control chronic LCMV infection, while being protected in a pristane-induced SLE disease model. Our study thus demonstrates the critical pathophysiological role of SLC15A4 and TASL/TASL2 for TLR7/9-driven inflammatory responses, further supporting the therapeutic potential of targeting this complex in SLE and related diseases.

Nucleic acid (NA) sensing by innate pattern recognition receptors (PRRs) is a central process to initiate immune responses against invading pathogens[1–3]. Recognition can occur via cytoplasm sensors, such as the cGAS-STING and RIG-I/MDA5-MAVS pathways, as well as by endolysosomal Toll-like receptors (TLRs), leading to the induction of antiviral and antimicrobial transcriptional programs and production of proinflammatory cytokines, chemokines and interferons[2–5]. While these responses are protective against infections, aberrant activation of these pathways is involved in several diseases, such as interferonopathies and NA-driven autoimmune conditions[6–8].

Endolysosomal NA-sensing TLRs comprise TLR3, 7 and TLR8 (in human) or TLR13 (in mice) which detect RNA and/or its degradation products, and TLR9, that responds to CpG-rich DNA[2,4,9]. Upon ligand engagement, these TLRs recruit the adaptor MyD88 (except for TLR3 which uses TRIF) to activate downstream NF-κB, MAPK and IRF pathways[2,4]. While IRF3 and IRF7 are the best known IRF transcription factors associated with PRRs, IRF5 is key for endolysosomal TLR7, 8 and 9 (TLR7-9) responses as it contributes to the induction of type I interferons (IFNs) as well as proinflammatory cytokines and chemokines[10,11].

Human and mouse studies strongly support the involvement of both the endosomal TLR7 and TLR9 (TLR7/9) pathways and IRF5 in the pathogenesis of SLE and related autoimmune conditions such as psoriasis and Sjögren's disease[12–19]. SLE is a complex and heterogenous

[1]Department of Immunobiology, University of Lausanne, Epalinges, Switzerland. [2]Lausanne Genomic Technologies Facility (LGTF), University of Lausanne, Lausanne, Switzerland. [3]Department of Pathology, Lausanne University Hospital (CHUV), Lausanne, Switzerland. ✉e-mail: manuele.rebsamen@unil.ch

disease, characterized by a loss of tolerance to NA, generation of autoantibodies and, in a large proportion of patients, a type I IFN signature[14]. Genome-wide association studies (GWAS) have identified over 80 genes, including TLR7, downstream signalling components and IRF5[17–22]. Indeed, IRF5 represents one of the strongest genetic factors associated with SLE and increased IRF5 activation has been reported in cells of SLE patients[10,23–25]. The recent identification of gain-of-function TLR7 mutations in SLE patients further supports this association[26]. In murine SLE models, increased or impaired TLR7 signalling leads to disease induction or protection, respectively, and IRF5 deficiency confers broad resistance[11,23,24,27]. Notably, the TLR7 pathway and IRF5 play a critical role not only in innate immune cells, including plasmacytoid dendritic cells (pDC), the main type I IFN producers, but also in B cells, influencing thereby production of pathogenic autoantibodies[7,15,28].

Among the genes associated with SLE by GWAS, SLC15A4, a lysosomal member of the proton-coupled oligopeptide transporter SLC15 family, has been shown to be required for endosomal TLR7-9 responses[17,21,22,29–38]. First evidence was provided by the identification of *Slc15a4*-mutant *feeble* mice in an in vivo ENU mutagenesis screen for impaired TLR7/9-induced IFN production[29]. Further studies showed that SLC15A4-deficient mice are unable to control chronic LCMV infection[39], while being strongly protected in a broad range of SLE models[31,40–43]. The mechanistic role of SLC15A4 in TLR7-9 responses remains in contrast poorly understood. Most studies propose that loss of SLC15A4 transport activity results in changes in lysosomal pH and/or content, altered cellular metabolism, and/or impaired mTORC1 activity affecting IRF7 regulation[31,33,34,44]. Moreover, SLC15A4 deficiency was also recently linked to impaired TLR9 trafficking and ligand engagement[42].

We recently discovered that SLC15A4 forms an IRF5-activating signalling complex with TASL, a previously uncharacterized protein encoded by another SLE-associated gene, *CXorf21*[17,38,45]. Mechanistically, TASL acts as an innate immune adaptor recruiting IRF5 through a C-terminal pLxIS motif, analogously to MAVS, STING and TRIF in IRF3 activation by their respective PRRs[38,46]. Notably, SLC15A4-mediated recruitment of TASL, but not SLC15A4 transport activity, is required for TLR7-9 responses as targeting TASL to the lysosomal compartment is sufficient to rescue IRF5 activation in SLC15A4-deficient human cell lines[47]. Supporting this, cryo-EM studies revealed that the N-terminus of TASL intrudes inside the central cavity of SLC15A4 in a cytoplasmic, inward open conformation, demonstrating that TASL recruitment is incompatible with SLC15A4 transport activity[48,49]. Lastly, we recently identified a first chemical inhibitor, *feeblin*, which locks SLC15A4 in a lysosomal outward open conformation incompatible with TASL binding, resulting in TASL degradation and the selective block of TLR7/8-induced responses in human cells[50]. While the discovery of the SLC15A4-TASL complex uncovered the signalling pathway linking TLR7-9 to IRF5 activation, the relevance of this complex in vivo remains unknown.

In this work, we show that loss of SLC15A4 blunts IRF5 activation in primary TLR7/9-responding cells. TASL knockout (TASL[KO]) leads to a profound but partial IRF5 impairment, which could be explained by the identification of a TASL paralogue encoded by *Gm6377*, TASL2. Accordingly, TASL[DKO] leads to compromised IRF5 activation and TLR7/9-induced transcriptional responses resulting in reduced production of proinflammatory cytokines and type I IFN, as well as IgG2c. Furthermore, TASL[DKO] phenocopies *feeble* mice in vivo, showing impaired viral control upon chronic LCMV infection, whilst being strongly resistant in a chemically induced SLE model. Altogether, these data reveal the central role of the SLC15A4-TASL-IRF5 pathway for endosomal nucleic acid sensing by TLR7 and TLR9, and highlights the SLC15A4-TASL complex as as potential therapeutic target for autoimmune diseases associated with aberrant activation of these receptors, such as SLE.

## Results

### SLC15A4 deficiency selectively blocks TLR7/9-induced IRF5 activation in pDC and B cells

Multiple mechanisms have been proposed to explain the defect in TLR7/9 responses observed in SLC15A4-deficient primary murine cells, but activation of IRF5 has not been investigated to date. To clarify this point, we first evaluated TLR7- and TLR9-induced responses in SLC15A4-deficient *feeble* mice[29]. In Flt3L bone marrow-derived pDC (BM-pDC), SLC15A4 deficiency abrogated IRF5 activation upon stimulation with the TLR7-agonist R848 (resiquimod), as assessed by IRF5 phosphorylation on Phos-tag-containing gels (Fig. 1a). In contrast, NF-κB and MAPK pathway activation, monitored by IκBα and JNK phosphorylation respectively, was largely unaltered (Fig. 1a). Similar selective impairment of IRF5 activation in *feeble* BM-pDC was observed upon TLR9 stimulation by CpG-B DNA (ODN1668) (Fig. 1b). Moreover, SLC15A4-deficient cells showed a moderate decrease in STAT1 phosphorylation, likely resulting from reduced paracrine IFN signalling (Fig. 1a, b). In line, IRF7 protein induction was diminished (Fig. 1c, d and Supplementary Fig. 1a–d). These data strongly suggest that loss of SLC15A4 in BM-pDC does not result in a major impairment of TLR7/9 engagement by the respective ligands, but specifically affects the downstream signalling pathway leading to IRF5 activation.

To further support these findings, we monitored splenic B cell responses. In line with previous reports[31,41], SLC15A4 deficiency strongly impaired TLR7- and TLR9-induced TNF and IL-6 production (Fig. 1e and Supplementary Fig. 1e, f), without affecting NF-κB nor MAPK activation (Fig. 1f, g). In contrast, IRF5 activation was blunted in *feeble* B cells, mirroring the data obtained in BM-pDC (Fig. 1f, g). While the defects in cytokine production in SLC15A4-knockout B cells have been previously linked to reduced mTOR activation and downstream IRF7 induction[31], we did not observe any major defects in S6 phosphorylation in B cells (Supplementary Fig. 1g). In BM-pDC, early S6 phosphorylation was similarly unaffected with a moderate reduction observed only at late timepoints, possibly secondary to early IRF5 defects (Supplementary Fig. 1h). Altogether, these data reveal that SLC15A4 is essential for TLR7/9-induced IRF5 activation in primary murine BM-pDC and B cells, suggesting that the phenotype previously observed in SLC15A4-deficient mice could be mechanistically linked to the SLC15A4-TASL-IRF5 pathway that we recently described in human cells[38].

### TASL deletion strongly impairs endosomal TLR7/9 responses

As the role of TASL in vivo remains unexplored, we generated *Tasl*-deficient mice by CRISPR-Cas9 mediated deletion of its single coding exon (Supplementary Fig. 2a–c). Murine *Tasl* (5430427O19Rik) is encoded on the X chromosome as its human counterpart, to whom it shares high amino acid sequence conservation, especially in the N-terminal region required for SLC15A4 binding and in the C-terminal IRF5-binding pLxIS motif (Supplementary Fig. 3b)[38]. TASL[KO] mice are viable, fertile and do not display any overt phenotype at steady state, nor any significant alteration in the myeloid and lymphoid immune compartments (Supplementary Fig. 2d).

We first assessed the effect of TASL loss in BM-pDC. Remarkably, TASL[KO] cells showed a profound defect in IRF5 activation upon R848 stimulation, even at high concentration, while displaying normal NF-κB and MAPK signalling (Fig. 2a and Supplementary Fig. 2e). TASL[KO] reduced STAT1 phosphorylation and IRF7 upregulation, suggesting a defect in IFNAR paracrine signalling (Fig. 2a, b and Supplementary Fig. 2e). Accordingly, R848-stimulated type I IFN induction was strongly affected by TASL deficiency (Fig. 2b, c). Similarly, TNF, IL-6 and IL-12p40 cytokine expression and secretion, while not abolished, was significantly reduced (Fig. 2b, c and Supplementary Fig. 2f). Importantly, TASL deficiency similarly impacted TLR7 responses in primary splenic pDC (CD11c[+], Siglec-H[+]) stimulated ex vivo. Indeed, type I IFN expression was abrogated in TASL[KO] cells and induction of

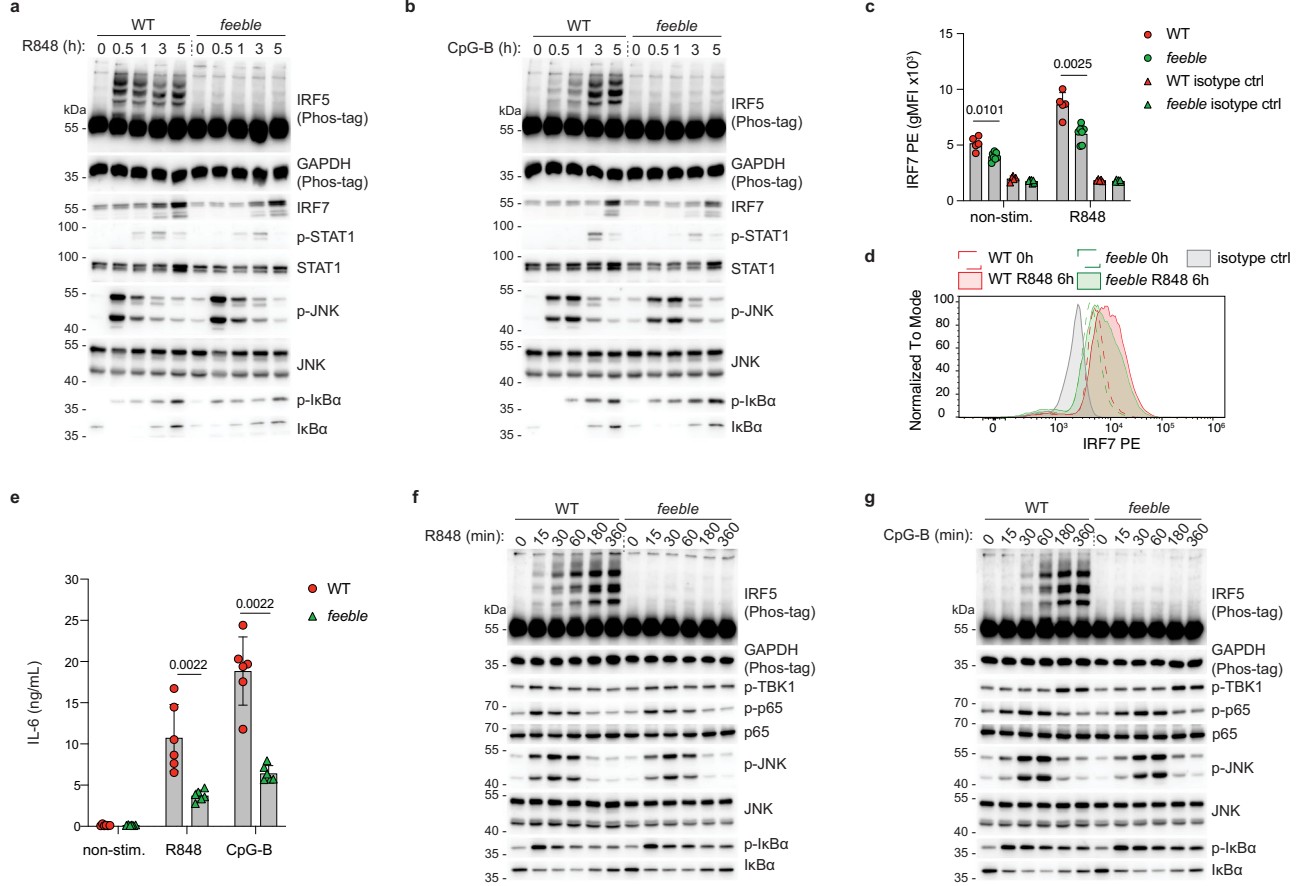

**Fig. 1 | SLC15A4 controls endosomal TLR7/9-induced IRF5 activation. a, b** WT and *feeble* female BM-pDC were stimulated with R848 (100 ng ml⁻¹) (**a**) or CpG-B (0.5 μM) (**b**) for the indicated time and lysates analyzed by immunoblotting. **c, d** Quantification of IRF7 induction in BM-pDC after R848 (6 h, 100 ng ml⁻¹) treatment measured by intracellular staining and quantified by gMFI (**c**) with representative histogram (**d**). Mean ± s.d. (WT n = 5 (3/2) and *feeble n* = 7 (4/3)). **e** IL-6 production of splenic B cells stimulated with R848 (100 ng ml⁻¹) or CpG-B (0.5 μM) for 24 h. Mean ± s.d. (WT *n* = 6 (0/6) and feeble *n* = 6 (0/6). **f, g** WT and *feeble* male splenic B cells were stimulated with R848 (100 ng ml⁻¹) (**f**) or CpG-B (0.5 μM) (**g**) for the indicated time and lysates analyzed by immunoblotting. Analysed by two-sided Mann-Whitney test (**c, e**). In (**a, b, f** and **g**) data are representative of at least two independent experiments. n=total number (males/females).

proinflammatory cytokines was strongly compromised, a defect which appeared more pronounced than in BM-pDC (Fig. 2d–f and Supplementary Fig. 2g–i). Notably, loss of TASL also strongly impaired pDC responses to TLR9 agonist CpG-B (Fig. 2g and Supplementary Fig. 2g–i).

Next, we investigated the impact of TASL deficiency in splenic B cells. Analogously to SLC15A4-deficient cells (Supplementary Fig. 1a, b), TASL^KO showed reduced IL-6 and TNF production upon R848 or CpG-B stimulation (Supplementary Fig. 2j). Surprisingly, while MAPK and NF-κB signalling was normal as expected, we observed that IRF5 activation was strongly reduced but not completely impaired (Supplementary Fig. 2k, l). This was not specific to B cells, as IRF5 activation upon TLR9 stimulation was still detectable also in TASL^KO BM-pDC (Supplementary Fig. 2m). The incomplete block of IRF5 activation in TASL^KO contrasts with the results obtained in *feeble* cells, in which IRF5 phosphorylation was virtually undetectable (Fig. 2h, i). This finding was unexpected considering that in the human system TASL deficiency fully phenocopied SLC15A4 loss[38,47], and suggested that either SLC15A4 has additional TASL-independent functions in endosomal TLR7/9 responses, or that other factor(s), partially redundant with TASL, exist in mice.

### *Gm6377* encodes a functional paralogue of TASL
Investigating this question, we identified a potential murine paralogue of *Tasl* encoded by *Gm6377* on the X chromosome. While *Gm6377* is absent in the human genome, orthologues are present in other vertebrates (Supplementary Fig. 3a). These proteins show amino acid sequence conservation in their N and C-terminal regions, which in human TASL mediate interaction with SLC15A4 and IRF5, respectively (Supplementary Fig. 3a, b)[38]. In the Immunological Genome Project (ImmGen) RNAseq database, *Gm6377* shows the highest levels in classical DC, but is broadly expressed among endosomal TLR7/9-responding immune cells (Supplementary Fig. 3c)[51]. We validated these data investigating primary splenic cells which demonstrated that both paralogues are indeed expressed in pDC, cDC, B cells and macrophages, albeit at different levels (Supplementary Fig. 3d). pDC displayed highest level of *Tasl* and lowest *Gm6377*, while cDC showed the opposite pattern. Given the functional homology with TASL described here below, we refer to GM6377 as TASL2.

To investigate if TASL2 shared functionality with TASL, we first assessed its ability to bind to SLC15A4. When co-expressed in HEK293T cells, murine SLC15A4 (mSLC15A4) co-immunoprecipitated TASL2 with an efficiency comparable to mouse and human TASL (m/hTASL) (Fig. 3a). Interestingly, while TASL and SLC15A4 interacted irrespectively of their human or murine origin, TASL2 specifically bound mSLC15A4, but not its human counterpart (Fig. 3a). Analogously to TASL, deletion of the first 8 N-terminal amino acids of TASL2 completely abrogated binding to mSLC15A4, whereas mutation in the pLxIS motif had no effect on complex formation (Fig. 3b). Next, we assessed TASL2 function in RAW 264.7 mouse macrophages, which rely on the SLC15A4-TASL complex to activate IRF5 upon R848 stimulation (Supplementary Fig. 3e). Stable overexpression of either mTASL

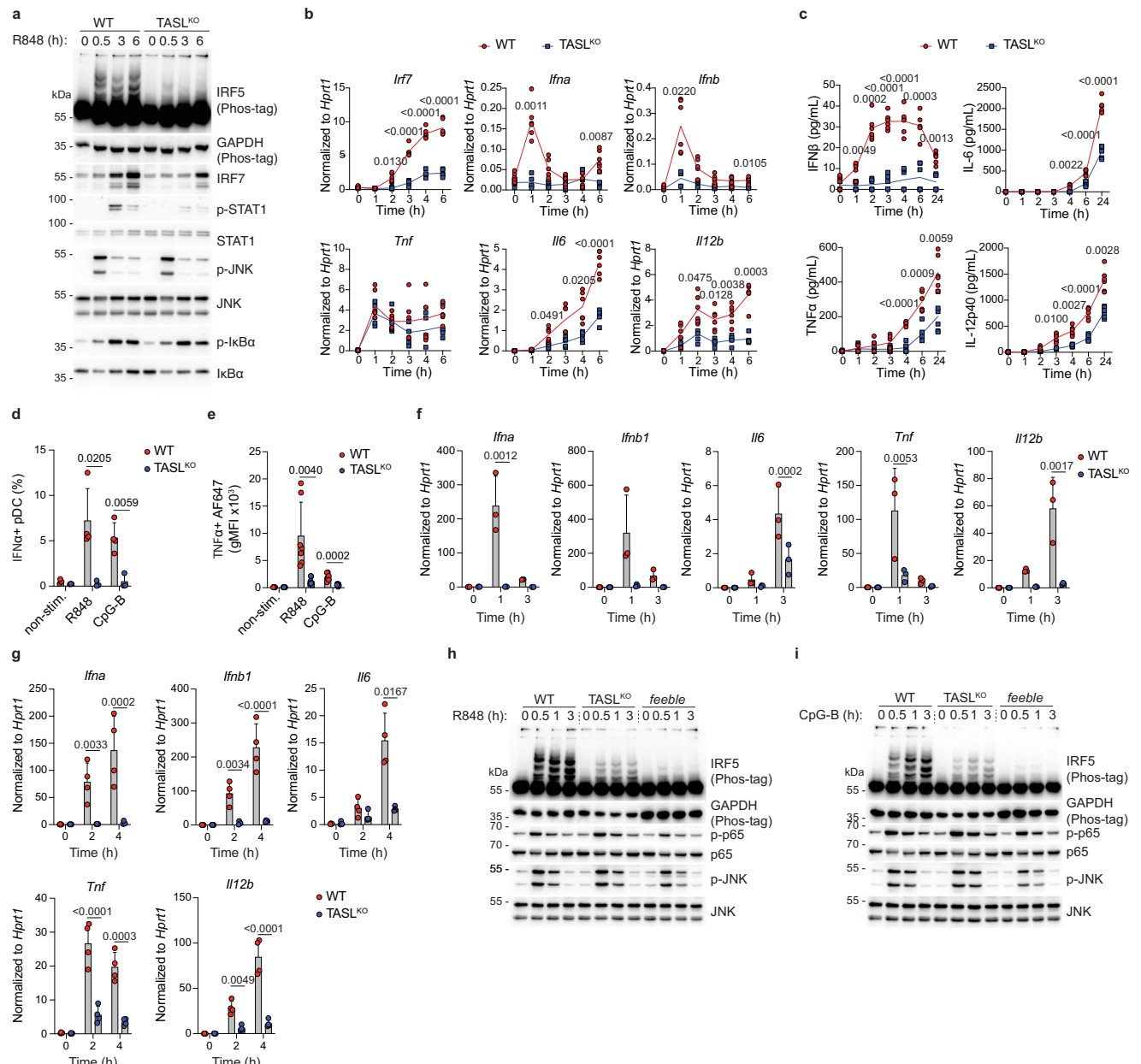

**Fig. 2 | TASL deficiency impairs IRF5 activation and downstream cytokine production. a** WT and TASL$^{KO}$ female BM-pDC were stimulated with R848 (100 ng ml$^{-1}$) and analyzed by immunoblotting. **b, c** BM-pDC were stimulated with R848 (100 ng ml$^{-1}$) for the indicated time and gene induction and cytokine production was analyzed by paired RT-qPCR (**b**) and ELISA (**c**). WT $n = 6$ (3/3), TASL$^{KO}$ $n = 5$ (2/3) for (**b**) and WT $n = 6$ (3/3), TASL$^{KO}$ $n = 6$ (3/3) for (**c**). Line represents mean. **d, e** Splenic pDC were activated with R848 (100 ng ml$^{-1}$) or CpG-B (0.5 µM) for 3 h in presence of Brefeldin A, intracellularly stained for IFNα (**d**) or TNFα (**e**) and analyzed by flow cytometry. WT $n = 4$ (0/4), TASL$^{KO}$ $n = 3$ (0/3) for (**d**) and WT $n = 7$ (4/3), TASL$^{KO}$ $n = 7$ (4/3) for (**e**). Mean ± s.d. **f, g** Sorted splenic pDC were treated with R848 (100 ng ml$^{-1}$) (**f**) or CpG-B (0.5 µM) (**g**) for the indicated time and analyzed by RT-qPCR. WT $n = 3$ (0/3), TASL$^{KO}$ $n = 3$ (0/3) for (**f**) and WT $n = 4$ (3/1), TASL$^{KO}$ $n = 4$ (3/1) for (**g**). Mean ± s.d. **h, i** Female splenic B cells from the indicated genotypes were treated with R848 (100 ng ml$^{-1}$) (**h**) or CpG-B (0.5 µM) (**i**) for the indicated time and analyzed by immunoblotting. Statistical analysis by two-way ANOVA with Šidák's multiple comparisons test (**b, c, f, g**) or two-sided unpaired $t$-test (**d, e**). In (**a, h** and **i**) data are representative of at least two independent experiments. $n =$ total number (males/females).

or TASL2 strongly promoted R848-induced IRF5 activation in wild-type (WT) cells (Supplementary Fig. 3f). Moreover, expression of TASL2 in TASL-deficient cells restored IRF5 activation similarly to mTASL, indicating that TASL2 can functionally substitute TASL (Fig. 3c). This activity required both the N-terminal, SLC15A4-binding region as well as the pLxIS motif (Fig. 3d). Of note, expression of TASL2 pLxIS mutant in control sg*Ren* RAW 264.7 cells appeared to further reduce IRF5 activation suggesting a dominant negative effect, possibly due to competition with endogenous mTASL for SLC15A4 binding (Supplementary Fig. 3g). This further suggested a similar mode-of-action

between mTASL and TASL2. Altogether, these data strongly support the notion that TASL2 is a functional paralogue of TASL, with the potential to compensate for TASL loss and contribute to the residual IRF5 activation we observed in TASL$^{KO}$ pDC and B cells.

We therefore generated TASL2-deficient mice by deleting its single protein coding exon (Supplementary Fig. 3h). TASL2$^{KO}$ splenic B cells showed no or minor reduction of IRF5 activation upon R848 or CpG-B stimulation, consistent with the dominant role of TASL we observed (Fig. 3e, f). We thus obtained double TASL and TASL2 knockout (TASL$^{DKO}$) by crossing. Similar to TASL$^{KO}$, TASL2$^{KO}$ or TASL$^{DKO}$

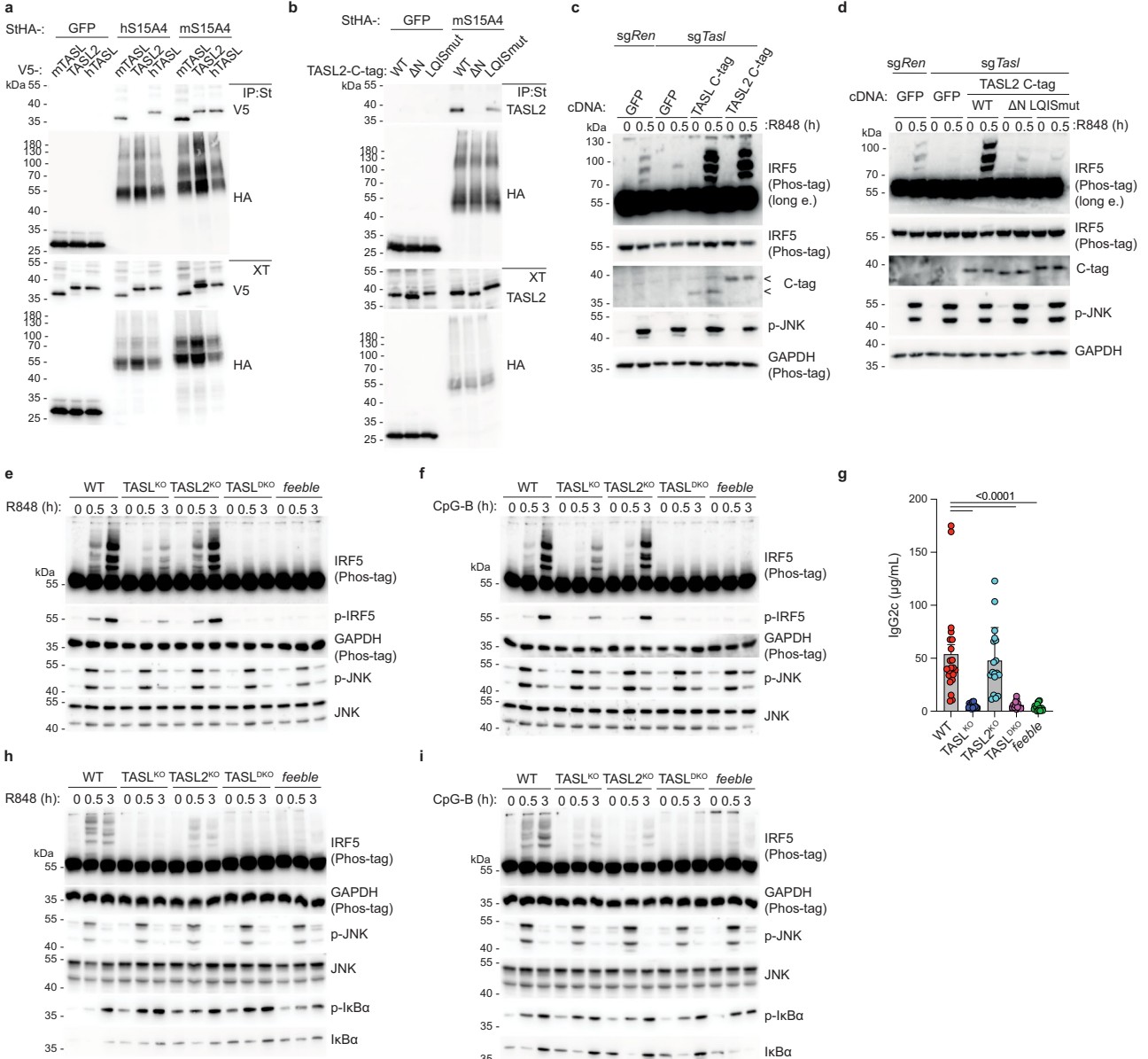

**Fig. 3 | *Gm6377* (TASL2) is a functional paralogue of *Tasl*.**
**a**, **b** Immunoprecipitates (IP, Strep-tag St) and whole-cell extracts (XT) from transiently transfected HEK293T cells were analyzed by immunoblotting. h, human; m, mouse; StHA, Strep-HA tag. **c**, **d** Immunoblots of control (sg*Ren*) or TASL knockout (sg*Tasl*) RAW 264.7 cell lines stably reconstituted with the indicated GFP, TASL or TASL2 constructs. Where indicated, cells were stimulated with R848 (10 µg ml⁻¹). **e**, **f** Female splenic B cells of indicated genotypes were treated with R848 (100 ng ml⁻¹) (**e**) or CpG-B (0.5 µM) (**f**) and analyzed by immunoblotting. **g** Steady

state serum level of IgG2c in individual genotypes. WT n = 22 (11/11), TASL^KO n = 16 (11/5), TASL2^KO n = 18 (10/8), TASL^DKO *n* = 13 (4/9), *feeble* n = 16 (4/12). Mean ± s.d. Analysis by one-way ANOVA with Dunnett's multiple comparisons test to WT. **h**, **i** Female BM-pDC of indicated genotypes were treated with R848 (100 ng ml⁻¹) (**h**) or CpG-B (0.5 µM) (**i**) and analyzed by immunoblotting. In (**a**–**f** and **h**, **i**) data are representative of at least two independent experiments. n=total number (males/females).

mice developed normally, and both young and aged animals did not display major defects in their immune compartment at steady state (Supplementary Fig. 3i–k). Remarkably, TASL double deficiency resulted in a complete impairment of IRF5 activation in splenic B cells irrespective of the endosomal TLR pathway triggered (Fig. 3e, f). Accordingly, TASL^DKO and *feeble* B cells showed a comparable reduction in IL-6 production, which largely correlated with the degree of residual IRF5 activation observed across the different genotypes (Supplementary Fig. 3l). SLC15A4 as well as IRF5 deficiency has been reported to strongly impact isotype switch resulting in specific reduction of IgG2a/c serum levels[27,31,32,52,53]. In line with this, TASL^KO and TASL^DKO displayed comparable impairment while TASL2^KO showed normal levels of IgG2c in serum (Fig. 3g). Finally, TASL^DKO phenocopied

the full block of IRF5 activation observed in *feeble* also in BM-pDC (Fig. 3h, i). These data strongly suggest that presence of either TASL or TASL2 is essential for TLR7/9-SLC15A4-dependent signalling, and that no other redundant pathway exists, at least in terms of IRF5 activation.

To further investigate the relative contribution of SLC15A4, TASL and TASL2 in a global and unbiased manner, we monitored transcriptional responses upon endosomal TLR stimulation across the five different genotypes (wild-type, *feeble*, TASL^KO, TASL2^KO and TASL^DKO). Transcriptional profiling (BRB-seq) was performed on BM-pDC unstimulated or treated with R848 or CpG-B, as well as on splenic B cells untreated or CpG-B stimulated (Supplementary Fig. 4a, b). As expected, in unstimulated conditions, SLC15A4, TASL and/or TASL2 deficiency did not have any major transcriptional effect in neither BM-pDC

nor B cells, with only a very limited number of significantly differentially expressed genes (DEG) detected across the different knockouts, including *Tasl* in the corresponding knockouts and a mild reduction of few interferon-stimulated genes (ISGs) in *feeble* BM-pDC, with similar trend detectable in TASL[DKO] (Supplementary Data 1, 2). In line with the altered IgG2c serum levels, its transcripts were the strongest downregulated in *feeble*, TASL[KO] and TASL[DKO] unstimulated B cells (Supplementary Fig. 4h). Analysis of R848- and CpG-induced responses in BM-pDC showed that TASL[DKO] substantially phenocopied SLC15A4 loss, with TASL[DKO] samples clustering with *feeble* in both conditions (Supplementary Fig. 4a, c). Accordingly, upon both TLR7 and TLR9 stimulation, the DEG profile in TASL[DKO] and *feeble* BM-pDC was largely comparable (Fig. 4a, b). The effect of single TASL[KO] or TASL2[KO] was less pronounced and dependent on the TLR triggered, with the number of DEGs correlating with the level of IRF5 impairment (Fig. 4a, b). Gene ontology analysis of DEGs downregulated in TASL[DKO] and *feeble* BM-pDC cells (compared to WT) identified significant deregulated processes fully consistent with impaired IFN and cytokine responses (such as "response to virus" and "IFN-mediated signalling pathway"), which was further highlighted by GSEA showing a clear downregulation of an ISG gene signature (Fig. 4c–e, Supplementary Fig. 4d–f and Supplementary Data 3–6). Supporting the data in BM-pDC, TASL[DKO] phenocopied *feeble* also in TLR9-stimulated B cells, with TASL[KO] having a stronger impact than TASL2[KO], correlating with their respective effect on IRF5 phosphorylation (Figs. 3f, 4f and Supplementary Fig. 4g).

Lastly, we assessed responses in cDC and macrophages, which express higher level of TASL2 (Supplementary Fig. 3d). In splenic cDC, TASL[DKO] impaired cytokine production to similar level as *feeble*, while single KOs showed only minor effect (Supplementary Fig. 5a). Similarly, TASL[DKO] phenocopied *feeble* in BM-derived cDC (BM-cDC), with TASL[KO] and TASL2[KO] showing partial and comparable reduction (Supplementary Fig. 5b). This suggests that in cDC TASL and TASL2 similarly contribute to responses and that each paralogue can largely compensate the loss of the other. Accordingly, TASL[DKO] blunted IRF5 activation in BM-cDC upon stimulation with either CpG-B or R848 (which triggered weak IRF5 phosphorylation in these cells) with TASL2[KO] showing comparatively stronger effect on IRF5 than TASL[KO] upon TLR9 stimulation (Supplementary Fig. 5c, d). Splenic macrophages showed similar impact of TASL[DKO] on cytokine production, which again phenocopied loss of SLC15A4. In these cells TASL[KO] also strongly reduced cytokine responses, while the effect of TASL2[KO] appeared to be more cytokine dependent (reduction of IL-12p40 and TNF, while no effect on IL-6) (Supplementary Fig. 5e).

Overall, these data indicate that TASL[DKO] is required to fully phenocopy SLC15A4 deficiency in terms of impaired IRF5 activation and downstream TLR7/9 induced responses, with single TASL[KO] and TASL2[KO] showing cell-type specific and partially redundant effects.

## Loss of TASL compromises responses to in vivo challenge with TLR7/9 agonists
To evaluate the relevance of TASL in TLR7/9 responses in vivo, we monitored serum levels of type I IFN, cytokines and chemokines upon intravenous challenge with R848 or CpG-B. Remarkably, IFNα and IFNβ were virtually undetectable in *feeble*, TASL[KO] and TASL[DKO] mice two hours after R848 stimulation, while other cytokines and chemokines were significantly reduced (Fig. 5a). TASL2[KO] had no or only marginal effect on these responses, largely correlating with the dominant effect of TASL observed in primary pDC and B cells ex vivo. Furthermore, profound defects were also observed upon CpG-B challenge, with a slightly increased impact of TASL2[KO], at least in terms of type I IFN production (Supplementary Fig. 6a). This effect was specific for endolysosomal NA-sensing TLR7/9, as IFNs and cytokine production upon LPS stimulation was largely unaffected (Supplementary Fig. 6b).

Next, we assessed the impact of impaired early TLR7/9-responses on downstream immune activation. Supporting a profound defect in type I IFNs, induction of ISGs was blunted in *feeble*, TASL[KO] and TASL[DKO] splenic pDC isolated 24 h after R848 or CpG-B injection (Fig. 5b and Supplementary Fig. 6c). Similar effect on ISGs induction was also observed in splenic cDC upon either of the treatments (Supplementary Fig. 6d, e). In line with this, activation of splenic pDC, cDC and B cells was strongly reduced (Fig. 5c and Supplementary Fig. 6f, g), as was bystander activation of CD4 and CD8 T cells, previously reported to be mediated by paracrine type I IFN signalling (Supplementary Fig. 6h)[54]. Altogether, these results demonstrate the crucial role of the SLC15A4-TASL complex for TLR7 and TLR9 responses in vivo. Indeed, even in acute settings induced by i.v. delivery of synthetic agonists, TASL[KO] and TASL[DKO] displayed blunted type I IFN responses and profound defects in cytokine and chemokine production, phenocopying SLC15A4-deficient *feeble* mice.

## *Feeble* and TASL[DKO], but not single TASL knockouts, fail to control chronic LCMV infection
Next, we investigated the relevance of the SLC15A4-TASL pathway in the context of LCMV cl.13 viral infection, whose long term control critically depends on TLR7 and was reported to be compromised in *feeble* mice[39,55]. 24 h after infection, serum levels of IFNs (type I and II), cytokines and chemokines were similarly reduced in *feeble*, TASL[KO] and TASL[DKO], with TASL2[KO] showing no major defect (Fig. 6a). To assess long term LCMV persistence, we monitored the presence of the virus in blood (viremia) over the course of 120 days. Interestingly, TASL[DKO], like *feeble*, displayed persistently circulating virus even at 4 months p.i. (Fig. 6b). In contrast, both TASL[KO] and TASL2[KO] were able to clear the virus from the blood (Fig. 6b). In line with blood titre, high viral RNA loads were detected in kidneys and lungs of *feeble* and TASL[DKO], while WT and single KOs showed comparable low levels (Fig. 6c). To investigate the causes of impaired viral control, we first monitored CD8+ T cell cytotoxic response at day 8 and 15 p.i. by assessing generation of effector cells (CD44+ CD62L−), KLRG1+ cytotoxic cells as well as upregulation of PD1, which did not reveal any major defects (Supplementary Fig. 7a–c). Furthermore, the magnitude of antigen specific CD8+ T cells, measured by MHC-I tetramers staining, was also unaltered (Supplementary Fig. 7d). In contrast, when we measured functional response after in vitro LCMV-peptides restimulation, we observed reduced number of double-cytokine producing cells (IFNγ+ TNF+) 8 days p.i. in all knockouts, with TASL[DKO] and *feeble* showing the strongest impairment compared to single TASL[KO] and TASL2[KO] (Fig. 6d and Supplementary Fig. 7e). Similar defect was present 3 months p.i. in both double and triple-cytokine producing cells (IFNγ+ TNF+ IL-2+) (Fig. 6e and Supplementary Fig. 7f, g). Therefore, TASL[DKO] mirrored the profound functional defect in LCMV-specific CD8 + T cells observed in *feeble*, suggesting its involvement in the inability to control the infection. Next, we investigated B cell responses. Interestingly, we could observe a partial impairment in the formation of short-lived plasma cells and germinal centre (GC) B cells in *feeble*, TASL[KO] and TASL[DKO] already detectable at day 8 and more pronounced at day 15 p.i. (Fig. 6f and Supplementary Fig. 7h–j). Accordingly, reduced GC B cell formation correlated with lower percentage of follicular helper T cells (Tfh) (Fig. 6g and Supplementary Fig. 7k). Lastly, we measured LCMV-specific IgG antibody titre at 120 days p.i. which showed the strongest reduction in *feeble* mice followed by TASL[DKO] and TASL[KO], while TASL2[KO] levels were same as WT (Fig. 6h). Altogether, these data revealed that the SLC15A4-TASL axis is critical to control chronic viral infection and that the two TASL paralogues are functionally redundant to restrict LCMV in vivo, as only TASL[DKO] show persistent viral titre as in *feeble* mice. Although TASL[KO] showed reduced immune responses, the clearance of the virus was only slightly delayed, indicating that residual TASL2 mediated activity is sufficient to provide protection.

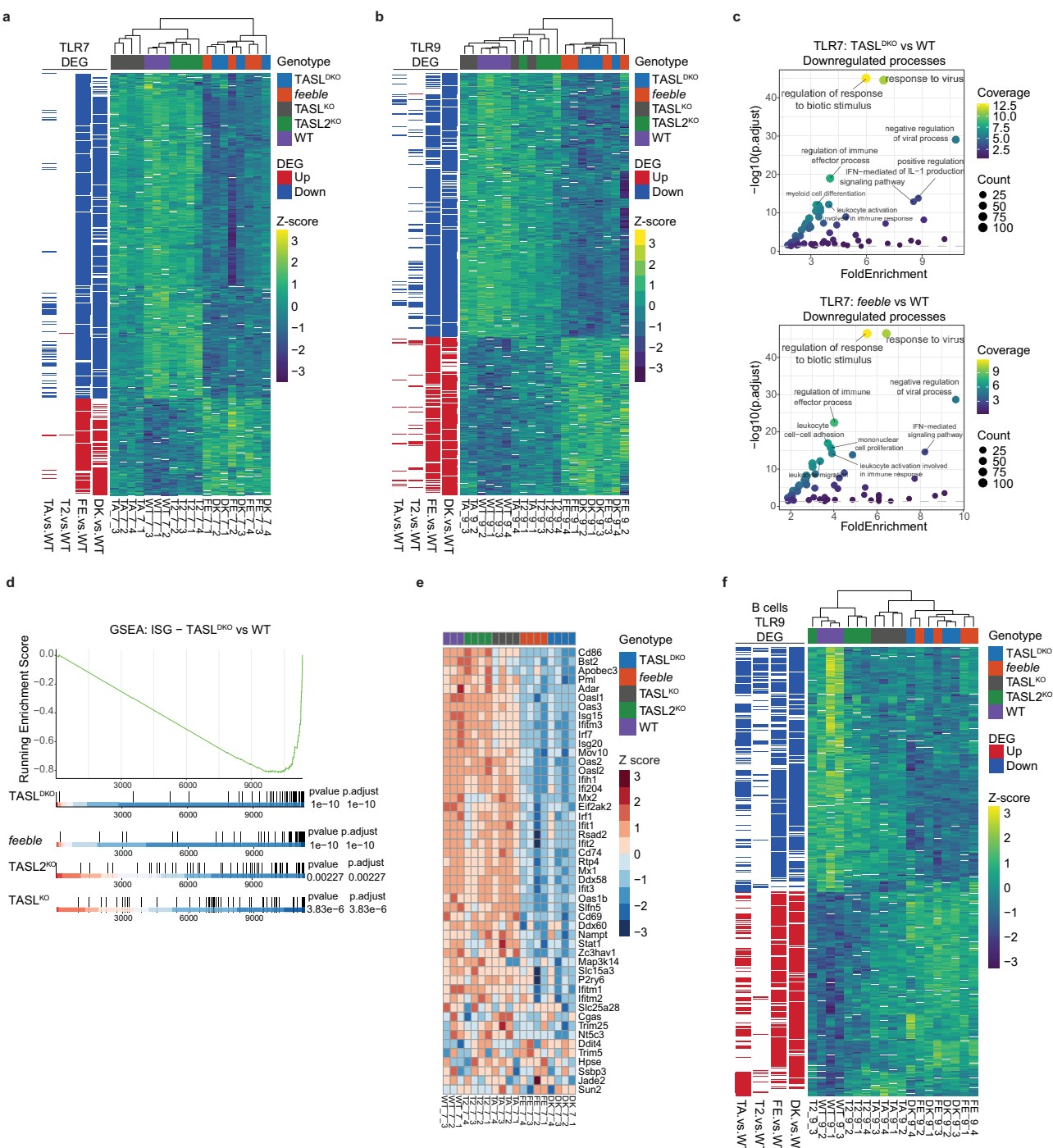

**Fig. 4 | TLR7/9-induced transcriptional responses in TASL^DKO phenocopy *feeble*.**
**a**, **b** Heatmap of identified differentially expressed genes (DEG) in BM-pDC from the indicated genotypes stimulated with R848 (100 ng ml⁻¹) for 4 h (**a**) or CpG-B (0.5 µM) for 6 h (**b**) compared to WT (adjusted $P < 0.05$). Colour intensity represents scaled expression levels. **c** Gene ontology (GO) enrichment analysis of biological processes for downregulated DEG in TASL^DKO (upper panel) or *feeble* (lower panel) BM-pDC after stimulation with R848 (100 ng ml⁻¹, 4 h) compared to WT. Each dot represents an enriched GO term (adjusted $P < 0.05$) with colour indicating the coverage and gene count determining the size. **d** Gene set enrichment analysis (GSEA) of an ISG signature in BM-pDC from the indicated genotypes after stimulation with R848 (100 ng ml⁻¹, 4 h) compared to WT. Gene lists were ranked by statistic. The upper panel depicts the running enrichment score specifically for

TASL^DKO, and the lower panels show the ranking of the ISG signature in each genotype and the related $P$ and adjusted $P$ values. **e** Heatmap of ISG signature genes in BM-pDC from the indicated genotypes after stimulation with R848 (100 ng ml⁻¹). Colour intensity represents scaled expression levels. **f** Heatmap of identified DEG in splenic B cells from the indicated genotypes stimulated with CpG-B (0.5 µM) for 6 h compared to WT (adjusted $P < 0.05$). Colour intensity represents scaled expression levels. Statistical analyses were performed using the two-sided Wald test from DESeq2 for differential expression analysis (**a**, **b**, **f**), one-sided Fisher's exact test for GO enrichment analysis (**c**), and two-sided permutation tests for GSEA (**d**). Adjustments for multiple comparisons were made using the Benjamini-Hochberg procedure. Female mice were used as a source of primary B cells and BM-pDC for transcriptional profiling.

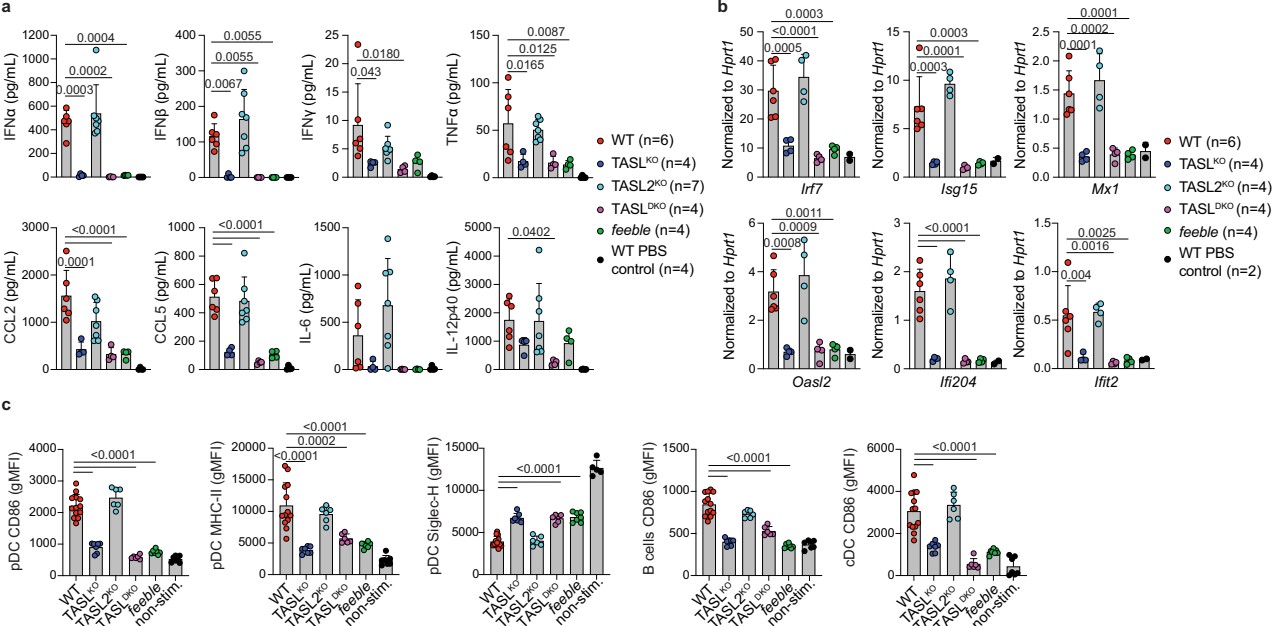

**Fig. 5 | TASL deficiency strongly impairs type I IFN and cytokine production upon in vivo challenge with TLR7/9 agonists. a** Serum levels of type I IFN, cytokines and chemokines 2 h after R848 (5 µg per mouse) i.v. challenge analyzed by ProcartaPlex assay. WT $n = 6$ (6/0), TASL^KO $n = 4$ (4/0), TASL2^KO $n = 7$ (4/3), TASL^DKO $n = 4$ (4/0), *feeble* $n = 4$ (4/0). Mean ± s.d. **b** Expression of the indicated ISGs in FACS-sorted splenic pDC 24 h after R848 i.v. challenge, analyzed by RT-qPCR. WT $n = 6$ (3/3), TASL^KO $n = 4$ (2/2), TASL2^KO $n = 4$ (3/1), TASL^DKO $n = 4$ (2/2), *feeble* $n = 4$

(2/2). Mean ± s.d. **c** Activation of indicated splenic cells 24 h after R848 injection measured by CD86, MHC-II or Siglec-H expression by flow cytometry (quantified as gMFI). WT $n = 13$ (6/7), TASL^KO $n = 7$ (3/4), TASL2^KO $n = 6$ (3/3), TASL^DKO $n = 6$ (3/3), *feeble* $n = 7$ (3/4), non-stim. $n = 7$ and $n = 5$ for Siglec-H gMFI. Mean ± s.d. Statistical analysis was performed with one-way ANOVA using Dunnet's multiple comparisons test compared to WT (**a–c**). n=total number (males/females). WT PBS control represents PBS injected mice.

## SLC15A4-TASL complex deficiency protects from chemically-induced lupus disease

Lastly, to assess the therapeutic potential of targeting the SLC15A4-TASL complex in autoimmune diseases, we investigated its relevance in a chemically-induced SLE model initiated by pristane (2,6,10,14-tetramethylpentadecane, TMDP) injection. We first monitored peritoneal immune cell infiltration 14 days after injection, focusing in particular on Ly6C^hi inflammatory monocytes, which in this model represent the main source of type I IFNs and correlate with the production of autoantibodies[56]. While the total numbers of peritoneal cells were comparable across genotypes, recruitment of Ly6C^hi monocytes was drastically impaired in TASL^KO and TASL^DKO mice, which phenocopied the effect of SLC15A4 deficiency (Fig. 7a and Supplementary Fig. 8a, b)[41]. In contrast, monocyte recruitment was largely unaffected in TASL2^KO. Importantly, IFN responses induced in wild-type animals and TASL2^KO were blunted in TASL^KO and TASL^DKO, as revealed by the impaired upregulation of ISGs in peritoneal infiltrates (Fig. 7b). These results demonstrate that the SLC15A4-TASL complex is essential to trigger early inflammatory responses. Next, we assessed how this translates to pathologic autoantibody induction 6 months after pristane injection by measuring major SLE-associated autoantigens. Serum levels of anti-Sm/RNP and anti-Su (Ago2) antibodies, targeting nuclear and cytoplasmic autoantigens respectively, were virtually blunted in *feeble*, TASL^KO and TASL^DKO (Fig. 7c). Similar impairment was also observed for nucleic acid targeting anti-RNA and anti-dsDNA autoantibodies (Fig. 7c). Accordingly, antinuclear antibodies (ANAs) were strongly reduced as shown by loss of the nuclear and cytoplasmic staining (Fig. 7d, e and Supplementary Fig. 8c). Finally, we assessed immune complex deposition in the kidneys, which, while not inducing significant glomerulonephritis in this model on C57Bl/6 J background[41,57], is one of the underlying causes of kidney damage in SLE patients[18]. Consistent with autoantibody levels, IgG deposition was reduced in TASL^KO and TASL^DKO mice, mirroring *feeble*

mice (Fig. 7f, g). Overall, these data demonstrate that the SLC15A4-TASL complex is critical for generation of disease-associated auto-antibodies and that TASL loss is sufficient to confer strong protection in this pristane-induced SLE model.

## Discussion

In this study, we show that TASL and its newly identified paralogue TASL2 play a central role in endosomal TLR7/9-induced responses in vivo, affecting both antiviral immune and autoimmune SLE models. While our recent discovery of TASL as an SLC15A4-associated innate immune adaptor uncovered a novel signalling complex mediating TLR7-9-induced IRF5 activation[38], the pathophysiological relevance of the SLC15A4-TASL-IRF5 pathway in vivo remained unexplored. Indeed, the phenotypes observed in SLC15A4-deficient mice have been previously ascribed to other mechanistic explanations, including impaired transport activity affecting lysosomal pH and histidine, altered cellular metabolism, defects in mTORC1 activation leading to impaired IRF7 induction as well as in endolysosomal trafficking preventing TLR-ligand engagement[30,31,33,34,42,44]. Assessing *feeble* mice, we show that SLC15A4 deficiency results in a specific block in IRF5 activation, which occurs early after TLR7 and TLR9 engagement, pointing to a primary defect. In addition to reduced proinflammatory cytokine production, *feeble* mutation affected downstream upregulation of IRF7 from IFN paracrine signalling, which could further contribute to dampen sustained IFN production. mTORC1 activation after TLR stimulation was largely unaffected, showing only a minor reduction in BM-pDC at later time point, suggesting a secondary effect due to early IRF5-dependent events.

These findings supported the generation of TASL-deficient mice. IRF5 activation in TASL^KO was strongly reduced but not blunted as in *feeble* cells, with the level of residual signal depending on the cell type and the TLR engaged. While initially surprising, we could explain these observations with the identification of a previously uncharacterized

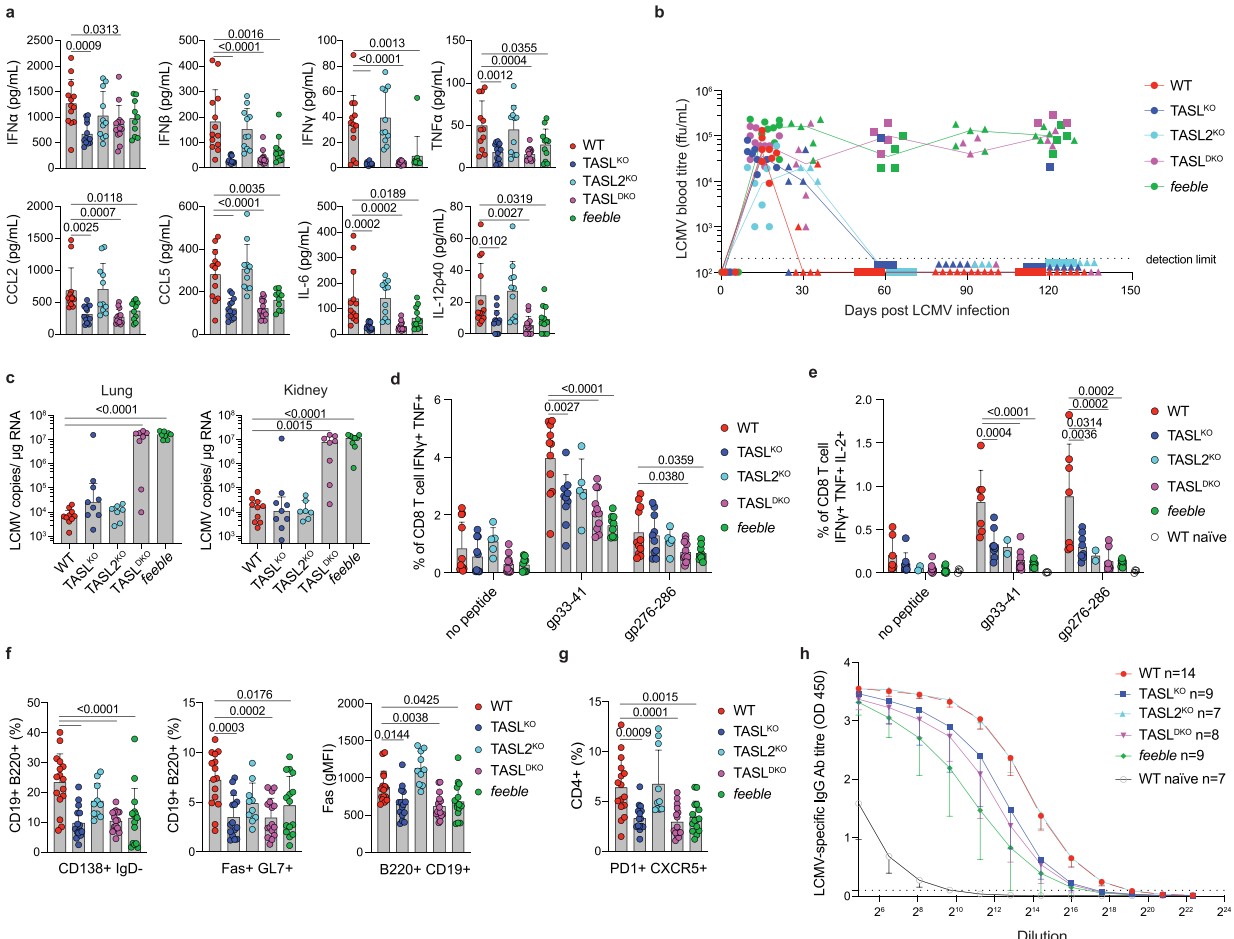

**Fig. 6 | TASL^DKO fail to clear chronic LCMV infection. a** Serum levels of type I IFN, cytokines and chemokines 24 h after LCMV infection analyzed by ProcartaPlex assay, WT *n* = 13 (2/11), TASL^KO *n* = 13 (2/11), TASL2^KO *n* = 11 (0/11), TASL^DKO *n* = 13 (5/8), *feeble n* = 10 (0/10). Mean ± s.d. **b** Blood viremia at indicated timepoints determined by focus forming assay. Combined data from 5 independent experiments, 3 exp. for day 15 (full circles, WT n = 8 (4/4), TASL^KO *n* = 8 (4/4), TASL2^KO *n* = 7 (3/4), TASL^DKO n = 10 (5/5), *feeble n* = 9 (2/7)) and 2 exp. for long term measurement up to 120 days (full triangles or squares respectively). Day 30 and 90: WT *n* = 6–10 (10/0), TASL^KO *n* = 5 (5/0), TASL2^KO *n* = 3 (3/0), TASL^DKO *n* = 5 (5/0), *feeble n* = 5 (5/0). Day 60: WT n = 4 (4/0), TASL^KO *n* = 4 (4/0), TASL2^KO *n* = 4 (4/0), TASL^DKO *n* = 4 (4/0), *feeble n* = 4 (4/0). Day 120 is endpoint for both long-term experiments. Lines indicate median. **c** Viral RNA load in organs 120 days after infection quantified by RT-qPCR. WT *n* = 10 (10/0), TASL^KO *n* = 9 (9/0), TASL2^KO *n* = 7 (7/0), TASL^DKO *n* = 8 (8/0), *feeble* n = 9 (9/0). Mean ± s.d. **d**, **e** Frequency of double (**d**) or triple (**e**) cytokine producing splenic CD8^+ T cells from 8 days (**d**) or 3 months (**e**) infected mice restimulated in vitro with indicated peptides for 5 h and analyzed by flow cytometry. WT *n* = 11 (6/5), TASL^KO *n* = 10 (5/5), TASL2^KO *n* = 5 (0/5), TASL^DKO *n* = 11 (6/5), *feeble n* = 9 (4/5) for (**d**) and WT *n* = 7 (7/0), TASL^KO *n* = 8 (8/0), TASL2^KO *n* = 2 (2/0), TASL^DKO *n* = 8 (8/0), *feeble n* = 7 (7/0) for (**e**). Mean ± s.d. **f**, **g** Frequency of short-lived plasma cells (CD138^+ IgD^−), germinal centre B cells (Fas^+ GL7^+) and level of Fas expression on splenic B cells (**f**) and frequency of follicular helper T cells in splenic CD4^+ cells (**g**) analyzed 15 days after infection by flow cytometry. WT *n* = 16 (4/12), TASL^KO *n* = 15 (4/11), TASL2^KO *n* = 10 (2/8), TASL^DKO *n* = 16 (6/10), *feeble n* = 15 (2/13). Mean ± s.d. **h** Antibody titre of LCMV specific total IgG in serum 120 days after infection. Mean ± s.d. Analysis performed with one-way ANOVA with Dunnet's multiple comparisons test compared to WT (**a**, **c**–**g**). n=total number (males/females). WT naïve group represent non-infected control mice.

paralogue, TASL2. Remarkably, TASL^DKO phenocopied *feeble* in all assays performed, i.e. ex vivo TLR7/9-induced signalling and global transcriptional responses, in vivo TLR7/9 stimulation as well as infection and autoimmune models. These data strongly indicate that SLC15A4 exerts its function via the TASL-IRF5 axis, and that other putative, TASL-independent functions of SLC15A4 are not essential for TLR7/9 responses, at least in the tested conditions. While the loss of SLC15A4 transport activity has been previously proposed to impair TLR7/9 signalling by altering endolysosomal pH and content, the central role of the SLC15A4-TASL complex in pathophysiological settings uncovered here is fully consistent with recent cellular and structural studies, which indicate that lysosomal docking of TASL is sufficient to rescue TLR7-9 responses in SLC15A4-deficient cells[47], and show that TASL binding to SLC15A4 is incompatible with its transport activity[48,49].

Interestingly, TASL and TASL2 are expressed in the different TLR7/9-responding cells we investigated, but their relative levels appear to be cell-type specific, showing opposite pattern in pDC compared to cDC. While future investigations will further detail the relative contribution of TASL and TASL2, we observed that TASL^KO has overall a more profound impact than TASL2^KO, both ex vivo and in vivo. TASL deficiency in primary pDC was sufficient to virtually abrogate type I IFN induction, which likely explains why TASL^KO mice showed a similar, profound defect in responses to TLR7/9 agonist injection as TASL^DKO. In cDC, which express the highest levels of TASL2, single TASL^KO had only a minor effect compared to its impact in other cell types, consistent with higher TASL2 levels translating in a stronger compensatory effect. Nevertheless, the impact of single TASL2^KO was similarly limited and TASL^DKO was required to phenocopy the

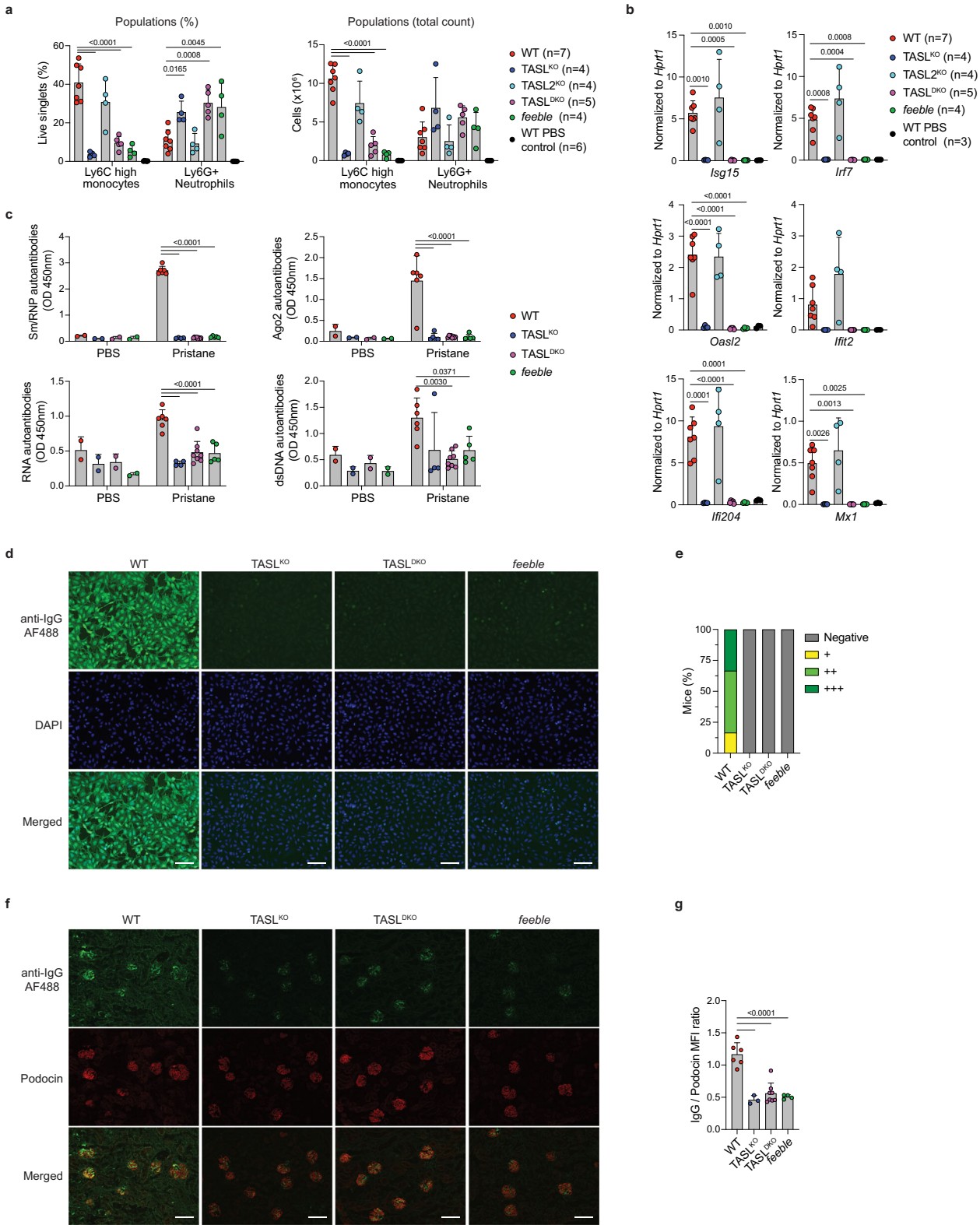

impairment observed in *feeble* cells. Interestingly, across our investigations we also noticed that TASL2 deficiency appear to have a relatively stronger impact on TLR9 responses than on TLR7. If confirmed, this is an interesting observation to be addressed in future studies. TASL[KO] had a greater effect than TASL2[KO] also in B cells, showing strongly reduced IgG2c serum levels as TASL[DKO], consistent with previous reports on SLC15A4-deficient mice and suggesting a defect in isotype switching[31,32]. This was further supported by transcriptional

analysis, which identified *Ighg2c* as one of the few and strongest downregulated transcripts in TASL[KO], TASL[DKO] and *feeble* in splenic B cells. Impairment of IgG2c in *feeble* and TASL[DKO] are consistent with reduced IRF5 activity. Indeed, IRF5 has been shown to be required for class switch recombination to IgG2a/c by controlling *γ2a/c* germ-line transcript expression[52]. Moreover, IRF5 affects two other important factors for class switch recombination by controlling TLR-induced T-bet upregulation and Ikaros downregulation[52,53]. Of note, TASL[KO],

**Fig. 7 | TASL$^{KO}$ and TASL$^{DKO}$ are protected in pristane induced autoimmunity. a** Flow cytometry analysis monitoring inflammatory monocytes and neutrophils in peritoneal cell infiltrates 14 days after pristane i.p. injection. WT $n = 7$ (7/0), TASL$^{KO}$ $n = 4$ (4/0), TASL2$^{KO}$ $n = 4$ (4/0), TASL$^{DKO}$ $n = 5$ (3/2), *feeble* n = 4 (4/0). Mean ± s.d. **b** Expression of selected ISGs in total peritoneal cells 14 days after pristane challenge, analyzed by RT-qPCR. WT $n = 7$ (7/0), TASL$^{KO}$ $n = 4$ (4/0), TASL2$^{KO}$ $n = 4$ (4/0), TASL$^{DKO}$ $n = 5$ (3/2), *feeble* n = 4 (4/0). Mean ± s.d. **c** Autoantibody levels against indicated autoantigens/nucleic acids in serum 6 months after pristane injection measured by ELISA. WT $n = 6$ (6/0), TASL$^{KO}$ $n = 4$ (4/0), TASL$^{DKO}$ $n = 8$ (8/0), *feeble* $n = 5$ (5/0). Mean ± s.d. **d**, **e** Antinuclear antibodies (ANA) in serum 6 months after

pristane treatment, assessed by Hep-2 immunofluorescence. Representative images for indicated genotypes (**d**) and evaluation of fluorescence intensity staining (**e**), WT $n = 6$ (6/0), TASL$^{KO}$ $n = 5$ (5/0), TASL$^{DKO}$ $n = 8$ (8/0), *feeble* $n = 5$ (5/0). Scale = 100 μm. **f, g** Immunofluorescence of histology kidney samples stained for IgG deposition in podocin-marked glomeruli. Representative images for indicated genotypes (**f**) with MFI ratio quantification (**g**) for each mouse are shown. Scale= 100 μm. WT $n = 6$ (6/0), TASL$^{KO}$ $n = 3$ (3/0), TASL$^{DKO}$ $n = 8$ (8/0), *feeble* $n = 4$ (4/0). One-way ANOVA with Dunnet's multiple comparisons test compared to WT (**a**–**c**, **g**). n=total number (males/females). WT PBS control represent PBS injected mice.

TASL$^{DKO}$ and *feeble* showed similar impaired regulation of these two factors as IRF5$^{KO}$ (Supplementary Fig. 4h). Therefore, impairment of IgG2c production appears, together with the reduction on cytokines induction by innate immune cells, as another important immunological consequence of SLC15A4-TASL complex deficiency.

The critical role of SLC15A4-TASL pathway was not restricted to challenge with synthetic agonists, as it was pivotal to control chronic LCMV infection, a viral infection model whose clearance is highly dependent on TLR7, rather than RIG-I-MAVS antiviral response[55,58]. TASL$^{DKO}$ mice lacked systemic viral control, and phenocopied *feeble*. In line with a previous study on *feeble* mice[39], this correlated with reduced early inflammatory cytokine production one day post infection. Assessing CD8+ T cells, we did not observe a major impairment in the extent of cytotoxic T cell responses nor in generation of LCMV antigen-specific cells across the KO lines. In contrast, we detected significant impairment in functionality, with a reduction in double (IFNγ$^{+}$ TNF$^{+}$) and triple-cytokine (IFNγ$^{+}$ TNF$^{+}$ IL-2$^{+}$) producing cells. TASL$^{DKO}$ fully phenocopied the profound defect observed in *feeble*, while single TASL$^{KO}$ and TASL2$^{KO}$ appeared to have a comparatively milder effect. In addition, we observed a defective humoral response, comprising reduced short-lived plasma cells and GC B cells together with a lower proportion of Tfh cells in the spleen, and significantly lower LCMV-specific antibody titre. Interestingly, both TASL$^{KO}$ and TASL2$^{KO}$ were able to efficiently resolve LCMV viremia with only minor delay compared to wild-type animals, supporting functional redundancy between these two paralogues in the context of chronic infection. These data suggest that residual IRF5 activity in single TASL knockouts is sufficient for virus clearance.

Lastly, we showed that TASL$^{KO}$ and TASL$^{DKO}$ were, as *feeble*, strongly protected in the pristane-induced SLE model, demonstrating the relevance of SLC15A4-TASL signalling complex for disease development. While the observation that TASL2$^{KO}$ had no impact on early inflammatory monocyte recruitment and ISG induction does not exclude its possible involvement in disease development at later stage, our data show that single TASL$^{KO}$ confer protection comparable to *feeble* and TASL$^{DKO}$. This contrasts with the TASL/TASL2 redundancy observed during LCMV infection and suggests that in this autoimmune model even a partial reduction in IRF5 activation is sufficient to prevent disease symptoms. This is consistent with previous reports showing that the haploinsufficiency of either *Slc15a4* or *Irf5* is largely protective in various SLE models[28,41,59]. Importantly, these data further suggest that partial pathway inhibition could be sufficient to yield therapeutic benefits in SLE, whilst potentially preserving antiviral immunity. In this regard, it should be noted that all current evidence indicates that the SLC15A4-TASL complex is specific for endolysosomal TLR7-9, while does not appear to be involved in the cytoplasmic nucleic acid-sensing pathways, such as RIG-like receptors or the cGAS/STING pathway. Therefore, it is expected that SLC15A4 and TASL deficiency or inhibition should preserve antimicrobial responses induced by these cytoplasmic pathways. Our ex vivo and in vivo data reveal that the SLC15A4-TASL complex is critical for two of the main pathogenic processes mediating SLE, i.e. IFN and proinflammatory cytokine production by innate immune cells and production of

autoantibodies by B cells. Together with the fact that the SLC15A4-TASL complex is druggable[50] and that full-body SLC15A4 and TASL-deficient animals do not display any overt phenotype at steady state, these findings strongly support the SLC15A4-TASL complex as a therapeutic target for SLE and related diseases. Future investigations should address whether and to what extent the SLC15A4-TASL pathway contributes to other TLR7/9- and IRF5-dependent responses and pathological conditions. Conversely, as in the case of viral infections, we do not expect the SLC15A4-TASL complex to be involved in TLR7/9- and IRF5-independent autoinflammatory and autoimmune models, such as the lupus-like manifestations observed upon TREX1-deficiency which relies on the STING pathway[60,61].

Collectively, the work presented here demonstrates the central pathophysiological relevance of the SLC15A4-TASL complex for TLR7/9- and IRF5-mediated proinflammatory, antiviral and autoimmune responses in vivo.

## Methods
### Mouse strains
C57Bl6/J strain (RRID:IMSR_JAX:000664) used as a WT control was originally obtained from Jackson Laboratory and bred and co-housed in the UNIL Epalinges animal facility, as for the other mouse lines, to obtain animals used in experiments. The *feeble* mouse strain used for this research project (C57BL/6J-Slc15a4$^{m1Btlr}$/Mmjax, RRID:MMRRC_034296-JAX) was obtained from the Mutant Mouse Resource and Research Centre (MMRRC) at The Jackson Laboratory, an NIH-funded strain repository, and was donated to the MMRRC by Bruce Beutler, Ph.D., The Scripps Research Institute. TASL and TASL2 knockout strains were generated by the Centre for Transgenic Models (CTM, Basel) by pronuclear injection of *Cas9* mRNA and gRNA (*Tasl*, atgggggctggtacaatagca, atggcaaactagaaagtcga; *Gm6377/Tasl2*, aacag ttcttccctagaaca, aaatatcaaaccaagctagt) into C57Bl6/J zygotes and surviving embryos transferred into pseudopregnant females. Resulting founders with deletion of entire coding sequence were backcrossed for at least 5 generations to WT C57Bl6/J background. For experiments, 6–12-week-old mice were used unless otherwise stated. Sex of mice is indicated in figure legends for each panel as well as in source data file for all individual data points. Mice were kept in individually ventilated cages in specific pathogen-free conditions with an irradiated standard rodent diet and water ad libitum in a facility with 12 h light/12 h dark cycle and stabile temperature 21 °C ( ± 1 °C) and 55 % (±10 %) humidity. Animals were euthanised by asphyxia using $CO_2$. All experiments involving animals were performed under the guidelines of and with approval from the cantonal veterinary office of the canton of Vaud (Switzerland), license number VD3716 and VD3779.

### Cell lines
HEK293T (CRL-3216) and RAW 264.7 (TIB-71) were purchased from ATCC. Cells were cultured in DMEM (Gibco, 10566016) supplemented with 10% (v/v) foetal bovine serum (FBS, Gibco) and antibiotics (100 U/ml penicillin, 100 μg/mL streptomycin, Bioconcept, 4-01F00-H) at 37 °C in 5% $CO_2$ incubator. Flt3L producing cells (B16 melanoma) were a kind gift from Prof. Steven Porcelli[62].

## Antibodies and reagents

Labelled antibodies CD11b Alexa Fluor 700 (1012229, Lot:B323579, dilution 1/200), CD11c Brilliant Violet 421 (117343, Lot:B318328, B347721, 1/200), Siglec-H AF647 (129608, Lot:B271031, 1/200), BST2 PE (127010, Lot:B344506, 1/200), F4/80 Alexa Fluor 488 (123120, Lot:B272102, 1/200), Ly6C PerCP-Cy5.5 (128011, Lot:B282010, 1/500), Ly6G Alexa Fluor 647 (127610, Lot:B255839, 1/200), MHCII (I-Ab) PE-Cy7 (116419, Lot:B289598, 1/400), CD8a Brilliant Violet 510 (100752, Lot:B336696, 1/200), B220 Brilliant Violet 711 (103255, Lot:B326247, B391673, 1/200), CD19 FITC (115521, Lot:B306929, 1/200), IgM Brilliant Violet 421 (405725, Lot:B293444, 1/200), IgD Alexa Fluor 700 (405729, Lot:B255981, 1/200), CXCR4 APC (146507, Lot:B278597, 1/200), CD138 PE (142503, Lot:B322376, 1/200), CD3e Alexa Fluor 488 (100321, Lot:B289380, 1/200), CD4 Brilliant Violet 421 (100437; Lot:B297643, B357846, 1/200), TCR-beta Brilliant Violet 785 (109249, Lot:B336011, 1/200), NK1.1 APC (108709, Lot:B316224, 1/200), CD44 PE (103007, Lot:B295248, 1/600), CD62L Alexa Fluor 700 (104426, Lot:B333915, 1/200), CD25 PE-Cy7 (102015, Lot:B290258, 1/200), KLRG1 APC (138412, Lot:B344824, 1/200), CD279 (PD1) PE-Cy7 (135215, Lot:B355883, 1/200), CD185 (CXCR5) BV605 (145513, Lot:B324823, 1/200), CD185 (CXCR5) PE (145503, Lot:B350818, 1/200), CD95 (Fas) PE-Cy7 (152617, Lot:B351840, 1/200), GL7 Alexa Fluor 647 (144606, Lot:B351840, 1/200), CD86 AF647 (105019, Lot:B308606, 1/200), CD86 AF488 (105018, Lot:B364517, 1/200), CD69 APC (104513, Lot:B361543, 1/200), TNFa AF647 (506314, Lot:B263283, 1/200 or 1/500 overnight), IL-6 (504503, Lot:B353935, 1/200), TCR-beta biotin (109204, Lot:B279070, 1/500), B220 biotin (103204, Lot:B335405, 1/500), Ly6G biotin (127604, Lot:B314607, 1/500), CD19 biotin (115504, Lot:B313073, 1/500), Isotype control IgG1, kappa, PE (RTK2071, Lot:B398003, 1/100), Isotype control IgG2a, kappa, Alexa Fluor 647 (RTK2758, Lot:B407447, 1/200) used in FACS were from BioLegend; C-tag biotin (7103252100, Lot:181002-01, 1/1000) was from ThermoFisher, Goat anti-Rabbit AF488 (A11034, Lot:2286890, 1/1000), goat anti-Rabbit Alexa Fluor 568 (A-11011, 1/1000), Podocin (PA5-79757, Lot:ZC4251691, 1/500), IRF-7 PE (12-5829-82, Lot:2410125, 1/100), phospho-IRF5 Ser437 (PA5-106093, Lot:YK4113295B, 1/1000) were from Invitrogen; Anti-mouse interferon Alpha (32100-1, Lot:7558, 1/50) was from PBL assay science. Antibodies against V5 (80076S), HA (3724), SAPK/JNK (9252), phospho-SAPK/JNK Thr183/Tyr185 (4668), NF-κB p65 (8242), phospho-NF-κB p65 Ser536 (3033), IκBα (4814), phospho-IκBα Ser32 (2859), STAT1 (14994), phospho-STAT1 Tyr701 (7649), phospho-TBK1/NAK Ser172 (5483), phospho-S6 (2215S) and IRF7 (72073) were from Cell Signalling and used at 1/1000 dilution; IRF5 (ab181553, Lot:GR3248905-4, GR3278824-6, 1/1000) was from Abcam; GAPDH (sc-365062, Lot:I2321, 1/1000) was from Santa Cruz. Biotinylated antibodies used in cells purification F4/80 biotin (130-116-514, Lot: 5221100468, 1/200), NK1 biotin (130-120-513, Lot:5221100457, 1/200), Siglec-H biotin (130-101-858, Lot:5220909807, 1/100) were from Miltenyi. Sigma-Aldrich (Merck): Cxorf21 (HPA001185, Lot:000030856, 1/500) Custom rabbit antibody against GM6377 (1/500) was generated with GenScript. Conjugated antibodies anti-mouse IgG HRP (115-035-003, Lot:161655, 1/5000), anti-rabbit IgG HRP (111-035-003, Lot:161658, 1/5000) used as secondary antibodies in Western Blot, goat anti-rat HRP (112-035-062, 1/2500) used in LCMV focus forming assay and anti-mouse IgG Alexa Fluor 488 (115-545-166, Lot:167698, 1/1000) used in microscopy were from Jackson ImmunoResearch. BioXCell: Rat anti-LCMV (BE0106, Lot:787521S1, 1/500). Strep-Tactin Sepharose resin (2-1201-002) was from IBA BioTAGnology. Brefeldin A (00-4506-51) was from Thermo-Fisher Scientific. R848 (tlrl-r848-5) and LPS (tlrl-3pelps) were from Invivogen; CpG-B ODN1668 was obtained from IDT. LIVE/DEAD™ Fixable Near-IR Dead Cell Stain Kit for FACS analysis and cell sorting was from Invitrogen (L34975). LCMV-specific peptides: GP33-41 H2-D(b) (H-KAVYNFATM-OH, GP276-286 H2-D(b) (H-SGVENPGGYCL-OH) were from peptides&elephants GmbH.

## Flow cytometry and cell sorting

The staining of live cells was performed in FACS buffer (PBS/2% FBS/2 mM EDTA) on ice usually for 30 min. For the phospho-specific staining, cells were fixed and permeabilized using Fix/perm kit (Biolegend, 420801, 421002). Samples were analysed on BD LSRII or Cytoflex S (Beckman Coulter) flow cytometer and data analysed using FlowJo (version 10.9.0, BD Biosciences). Gating strategies are illustrated in Supplementary Fig. 9 and Supplementary Fig. 10. Cell sorting was carried on BD FACS Aria (BD Biosciences) either with 85 or 100 μm noozle on MACS pre-enriched splenocytes.

## Cell purification, in vitro activation

Primary cells from spleen were isolated by gentle passing through 70 μm filter and erythrocytes lysed using ACK buffer. Splenic B cells were isolated using untouched magnetic separation (B cell isolation kit, Miltenyi, 130-090-862) on LS columns. For splenic pDC, cell suspension was enriched by magnetic depletion of CD19[+], TCRbeta[+] cells before sorting for CD11c[+] and Siglec-H[+] pDC while splenic cDC were sorted as CD11c[high] and Siglec-H negative (Supplementary Fig. 10c). Splenic macrophages were isolated using positive magnetic separation with F4/80 biotin antibody.

Bone marrow was flushed from femurs and tibiae and used for generation of bone marrow derived pDC (BM-pDC) in 5–10% Flt3L conditioned complete IMDM medium (10% FBS, 100 U/ml penicillin, 100 μg/mL streptomycin) differentiated for 8–9 days. At the end of differentiation, cells were measured by flow cytometry using CD11c[+] and Siglec-H[+] markers. BM-pDC were further isolated by positive magnetic separation using Siglec-H biotin and anti-biotin Microbeads (Miltenyi, 130-090-485).

MACS isolated B cells, BM-pDC and macrophages or sorted splenic pDC and cDC were resuspended in complete IMDM and rested in incubator for at least 30–60 min before stimulation with R848 or CpG-B. Concentrations of R848 and CpG-B are indicated in Figure legends for corresponding experiments. For cytokine intracellular staining, cells were activated either for 1 h (B cells) or 30 min (BM-pDC, splenic pDC) before addition of Brefeldin A (ThermoFisher Scientific, 00-4506-51) at the final concentration 3 μg/mL.

For LCMV peptide restimulation, total splenocytes from infected mice (day 8 or 90 post infection) were incubated with individual peptides (100 nM final concentration) or left without restimulation in the presence of Brefeldin (3 μg/mL) for 5 h before FACS analysis.

## Immunoprecipitation and immunoblotting

For immunoprecipitation experiments, HEK293T cells were transfected with the indicated plasmids encoding for tagged proteins using PEI (PolyScience, 23966). After 6 h, the media was exchanged to fresh one and after 24 h, the cells were lysed using E1A buffer (50 mM Hepes pH 7.4, 250 mM NaCl, 5 mM EDTA, 1% NP40) complemented with phosphatase (Thermo Fisher Scientific, 78420) and protease (Roche, 11836170001) inhibitors. Whole cell lysate sample was removed as input, the rest subjected to immunoprecipitation using equilibrated anti-Strep-Tactin Sepharose beads (IBA BioTAGnology, 2-1201-002) and incubated over night at 4 °C with rotation. Beads were washed three times with E1A buffer and eluted with 5 mM biotin (Sigma, B4501). Whole cell lysate and immunoprecipitated proteins were treated with PNGase F (BioLabs, P0704L) for 30 min at 37 °C before addition of SDS sample buffer, then analysed by SDS-PAGE and immunoblotting.

For immunoblotting, cells were lysed in RIPA lysis buffer (25 mM Tris, 150 mM NaCl, 0.5% NP-40, 0.5% deoxycholate (w/v) and 0.1% SDS (w/v), pH 7.4) supplemented with Roche EDTA-free protease inhibitor cocktail (one tablet for 50 mL), Halt phosphatase inhibitor cocktail (Thermo Fisher Scientific,78420) and Benzonase (Merck, 71205), 10 min on ice. Lysates were cleared by centrifugation (18,000 g, 10 min, 4 °C) and proteins quantified with BCA (ThermoFisher

Scientific) using BSA as standard. Cell lysates were resolved by regular or Phos-tag-containing (20 μM, WAKO Chemicals, WA3 304-93521) SDS−PAGE. After electrophoresis, Phos-tag-containing SDS−PAGE were soaked in transfer buffer with 10 mM EDTA for 3 × 10 min, rinsed 10 min in transfer buffer without EDTA and blotted to nitrocellulose membranes (Amersham, Glattbrugg, Switzerland). Membranes were blocked with 5% non-fat dry milk in TBST and probed with the indicated antibodies. Binding was detected with anti-mouse-HRP secondary antibodies (115-035-003) or anti-rabbit-HRP secondary antibodies (111-035-003), from Jackson ImmunoReasearch, using the ECL western blotting system (Advansta, K-12045-D50). When multiple antibodies were used, equal amounts of samples were loaded on multiple SDS−PAGE gels and western blots were sequentially probed with a maximum of two antibodies.

## Plasmid constructs

Codon-optimized cDNAs for human SLC15A4 and TASL were obtained from Genscript[38]. cDNA for mouse SLC15A4, TASL and TASL2 were also obtained from Genscript and subcloned into pDONR201 or pDONR221 (Invitrogen) plasmids. TASL ΔN (2−8) and TASL LQIS > AQAA constructs were generated by site-directed mutagenesis using the PfuUltra High-Fidelity DNA Polymerase AD (Agilent). C-tag (encoded by amino acids EPEA)[63] was added at the C-terminus of the different constructs by site-directed mutagenesis. All cDNAs were verified by sequencing and shuttled to a pRRL-based lentiviral expression plasmids with a selectable resistance cassette, allowing Ctag, N- or C-terminal Strep-HA-tagged (StHA) or V5-tagged expression. Reconstitution experiments were performed using codon-optimized cDNA resistant to sgRNAs targeting the endogenous genes.

CRISPR–Cas9-based knockout cell lines were generated by viral transduction of pLentiCRISPRv2 (Addgene plasmid no. 52961) containing the sgRNA sequence and a puromycin selection marker. Lentiviral packaging plasmids psPAX2 and pMD2.G were obtained from Addgene (plasmid no. 12260 and 12259). The oligonucleotides for the control sgRNA targeting Renilla luciferase (sgRen) are: forward F: CACCGGTA-TAATACACCGCGCTAC, R: AAACGTAGCGCGGTGTATTATACC[64]. The other sgRNAs were designed using the Broad Institute sgRNA design tool (https://portals.broadinstitute.org/gpp/public/analysis-tools/sgrna-design)[65]. Cloned oligonucleotide sequences are provided in Supplementary Table 1.

## Lentiviral transduction

For lentiviral gene transduction, HEK293T cells were transfected with the respective lentiviral vectors and packaging plasmids psPAX2 and pMD2.G using PEI (PolySciences, 23966). Six hours later, medium was exchanged with fresh complete DMEM, (10% FBS (v/v), 100 U/ml penicillin, 100 μg/ml streptomycin). Forty-eight hours after transfection, virus-containing supernatants were collected, filtered through 0.45 μm polyethersulfone filters (Millipore) and supplemented with 5 μg/ml polybrene (Sigma, H9268). Cells were infected by spin infection (800 g, 45 min room temperature). Twenty-four hours after infection, cells were washed and resuspended in fresh medium containing appropriate antibiotics (puromycin or blasticidin). Selected cell populations were used for experimental investigations without further subcloning to avoid clonal effects.

## Transcriptional profiling by BRB-Seq

**Library preparation and sequencing.** RNA libraries were prepared using BRB-Seq kit from Alithea Genomics, according to the manufacturer's recommendations. A total of 85 ng purified RNA was transferred into the BRB-Seq multiwell plate containing the barcoded oligodT primers for the reverse transcription step. All cDNA were pooled into an Eppendorf tube and purified using SPRIselect beads (Beckman) at a 1X ratio. The remaining free primers were digested with an exonuclease step. Second strand synthesis was performed from the purified cDNA

pool and cleaned-up with a 0.6X SPRIselect beads step. Double-stranded cDNA was then tagmented and purified with a 0.6X SPRIselect beads step. Finally, the 3' terminal fragments were amplified using Unique Dual Indexing (UDI) Illumina adapter primers. After a final 0.7X SPRIselect beads purification step, the library generated was quantified with a fluorimetric method (QubIT, ThermoFisher Scientific) and its size pattern was analysed with a fragment analyzer (Agilent). Sequencing was performed on an Illumina NovaSeq 6000 with 28-8-8-91 cycles (read1 - index i7 - index i5 - read2). Sequencing data were demultiplexed using the bcl2fastq2 Conversion Software (version 2.20, Illumina).

**Data processing and differential expression.** The fastq file was demultiplexed using BRBseq Tools software (version 1.6)[66]. Subsequently, the RNA-seq pipeline was executed on each demultiplexed Fastq file, following these steps: Firstly, reads were subjected to adapter and quality trimming using Cutadapt (version 2.5)[67]. Next, reads corresponding to ribosomal RNA sequences were eliminated using fastq_screen (version 0.11.1)[68] The remaining reads underwent further filtering for low complexity with reaper (version 15-065)[69]. These processed reads were then aligned to the *Mus musculus* genome (build GRCm38) using STAR (version 2.5.3a)[70]. UMI deduplication was performed after the mapping step using UMI-tools dedup functionality (version 0.5.3)[71]. The count of unique UMIs per gene locus was summarized using htseq-count (version 0.9.1)[72] with Ensembl 102 gene annotation. The quality of the RNA-seq data alignment was evaluated using RSeQC (version 2.3.7)[73].

The statistical analysis for genes was conducted in R (version 4.3.0). B-cells and bone marrow pDC datasets were independently analysed through the following steps:

First, genes with low counts were filtered out based on the criterion of having at least 3 counts per million (cpm) for B-cells dataset and 2 cpm for bone marrow pDC dataset, in at least 30% of samples. Sample exclusion was applied to samples with less than 600'000 reads aligned for B-cell dataset and 3.5 million aligned reads for the pDC dataset. The sample WT_9_4 in the B-cell dataset exhibited an outlier behaviour and was excluded from the analysis. Subsequently, normalization and differential expression analysis were performed using DESeq2 (version 1.40.1)[74], with default settings. Samples from each independent dataset were fitted together, and the statistical significance was computed using the Wald statistic. For each comparison, P-value adjustment was carried out using the Benjamini-Hochberg method to control the false discovery rate (FDR). Genes with a significant 5% FDR cutoff were selected. Hierarchical clustering utilised the correlation distance with the complete method, and heatmaps were generated using pheatmap function (version 1.0.12)[75]. For over-representation analysis (ORA) of significant genes, Gene Ontology terms related to biological processes (GO BP, org.Mm.eg.db version 3.17.0) were employed. This analysis was performed using the R package clusterProfiler (version 4.8.1 enrichGO function)[76]. P-values were calculated with a one-sided Fisher's exact test and adjusted for multiple testing using the Benjamini-Hochberg method. Separate analyses were conducted for up-regulated and down-regulated gene lists. The result list of enriched gene sets was further simplified to remove redundant GO terms using the simplify function from the same package, with the parameters cutoff=0.5. Gene Set Enrichment Analysis (GSEA)[77] was carried out using the clusterProfiler package on the ISG signature[78,79]. Each Wald statistic-ranked gene list served as input for GSEA with default parameters, determining whether the defined gene sets were randomly distributed throughout the ranked gene list or predominantly found at the top or bottom. The p-value for each enrichment score (ES) was calculated by permutation test, and multiple testing adjustment was performed using the Benjamini-Hochberg method.

## In vivo TLR agonist treatment

Mice were injected intravenously either with R848 5 μg (Invivogen tlrl-r848-5), CpG-B 5 μg (ODN1668, IDT) or LPS 20 μg (Invivogen, tlrl-

3pelps) in 200 μl of PBS. For CpG-B injection per one mouse, 5 μg of ODN1668 and 8 μl of DOTAP (Roche, 11202375001) were resuspended in separate Eppendorf tubes in 50 μl of HBSS each and then mixed and incubated for 15 min. After incubation, HBSS was added up to 200 μl and immediately used for injection.

### Cytokine and chemokine profiling

Serum samples from mice were obtained by coagulation of blood from retroorbital sinus at room temperature for 20–30 min followed by centrifugation at 2000g for 20 min. Profiling of cytokines in serum was performed by custom-prepared ProcartaPlex (ThermoFisher, PPX-08-MXXGTT3) following manufacturer's protocol. Beads were analysed on Bio-Plex 200 system (Biorad). Following cytokines and chemokines were detected: IFNα, IFNβ, IFNγ, IL-12/IL-23p40, IL-6, MCP-1 (CCL2), RANTES (CCL5), TNFα.

### ELISA

ELISA experiments were carried out using diluted cell supernatants or mice serum and multiple cytokines were measured according to the manufacturer's instructions. ELISA kits for mouse IgG2c (88-50670-22), mouse TNFα (88-7324-77), mouse IL-6 (88-7064-88) and mouse IL-12p40 (88-7120-88) were purchased from Invitrogen. Mouse IFNβ was determined by DuoSet ELISA (R&D Systems, DY8234-05). Autoantibody detection was done using ELISA plates (ThermoFisher Scientific, 439454) coated with RNP/Sm antigen (2,5 μg/mL, Arotec diagnostics, ATR01) or Ago2 (1 μg/mL, SinoBiological, 50683-M07B). For nucleic acid autoantibody, plates were precoated with poly-L-Lysine (50 μg/mL, Merck, P6282) and then coated overnight with either RNA (ThermoFisher Scientific, AM7120G) or DNA (15633019) at the concentration of 2,5 μg/mL.

### RT-qPCR

RNA was isolated either using RNeasy Plus micro kit (Qiagen, 74034) or Quick-RNA Miniprep plus kit (ZymoResearch, R1058). RevertAid First strand cDNA Synthesis Kit (Thermo Fisher Scientific, K1621) was used for reverse transcription. Mouse specific DNA primers used in this study were purchased from Sigma-Aldrich and their sequences are listed in Supplementary Table 1.

### LCMV infection and viral quantification

LCMV clone 13 was obtained from S. Luther (Department of Immunobiology, UNIL). Virus was produced by infection of hamster BHK-21 cells (MOI = 0.01) and collection of supernatant 48 h later. The concentration of virus was determined by focus forming assay on 3T3 cells[80] using anti-LCMV antibody (BioXCell, BE0106) on cells infected with serial dilution of viral supernatant. Mice were infected by intravenous injection of $2 \times 10^6$ FFU. Quantification of circulating virus was done on serial dilution of full blood by focus forming assay. Viral load in organs was assessed by quantitative PCR with reverse transcription (RT-qPCR). Total RNA from homogenized organs was isolated by TRIzol LS (Invitrogen, 10296010) and RNA Clean & Concentration kit (ZymoResearch, R1014) with in-column DNase treatment. RNA was transcribed using RevertAid First strand cDNA Synthesis Kit (ThermoFisher Scientific, K1621) with oligo(dT)18 and LCMV-specific cDNA primers (Supplementary Table 1). RT–qPCR was performed using KAPA SYBR FAST (Roche, KK4611) and a LightCycler 480 II (Roche). Cloned S-segment of LCMV (in pBlueScript II) was used as a standard (10x serial dilution) for absolute quantification of viral RNA load.

### Pristane injection

Mice for both, short term (14 days) and long-term (6 months) experiments were injected with 0.5 mL of Pristane (2,6,10,14-Tetramethylpentadecane, TMPD; Merck P2870) intraperitoneally. Infiltrating cells were isolated by peritoneal lavage with prewarmed (37 °C) FACS buffer (12 mL) and aspiration of 10 mL. Cell count was normalized to 12 mL of injected buffer.

### Histology

For histology imaging, kidneys were harvested and embedded in Tissue-Tek OCT Compound (Sakura, 4583) and frozen. After cutting, sections were fixed with acetone 100% for 10 min at 4 °C and rehydrated in PBS 1X three times for 5 min, with a change of PBS between each time. Sections were blocked with assay mix (PBS 1X, 3% BSA, 0.1% Triton X-100 (Sigma, 93426)) for 2 h at RT in a humid box, then stained with a rabbit anti-podocin primary antibody (Invitrogen, PA5-79757) diluted 1/500 in assay mix for 45 min at RT. Slides were washed 3 times with PBS 1X. Secondary antibodies goat-a-rabbit AF568 (Invitrogen, A-11011) and goat anti-mouse IgG Alexa Fluor 488 (Jackson ImmunoResearch, 115-545-166) were added on section and incubated 45 min at RT in a humid box. Slides were washed 3 times in PSB 1X and stained with DAPI (Invitrogen, R37606). Three other washes with PBS 1X were performed and slides were mounted with coverslips using a Fluoroshield mounting medium (Sigma, F6182). Images were acquired using Zeiss Upright AxioVision. IgG deposition in glomeruli was quantified by mean fluorescence intensity staining by selecting identical area sizes over 3 glomeruli per image and 2 images per mouse. MFI ration (IgG/Podocin) is used to visualize the data and mean value (from 6 glomeruli in total) per mouse is used for plotting in the graph.

### ANA staining

Antinuclear antibodies assay was performed on mouse serum (1/300 dilution) using Hep-2 ANA kit (NOVA Lite, 708100) and following the manufacturer's instructions. The only change was the replacement of goat anti-human IgG (in kit) with goat anti-mouse IgG Alexa Fluor 488 (Jackson ImmunoResearch, 115-545-166). All images were acquired using Zeiss Upright AxioVision with the same setting.

### Statistical analysis

No statistical methods were used to predetermine sample size. Sample sizes and experimental animal groups were estimated based on experiences with similar experiments done by us or by in published studies. Methods of statistical analysis used to determine the significance of difference are listed in corresponding figure legends, and the exact P values are indicated in figures. Statistical analyses were performed using the Prism software (GraphPad).

### Reporting summary

Further information on research design is available in the Nature Portfolio Reporting Summary linked to this article.

## Data availability

Primary BRB-seq data are available under GEO accession number GSE267786. All data are included in the Supplementary Information or available from the authors, as are unique reagents used in this Article. The raw numbers for charts and graphs are available in the Source Data file whenever possible. Source data are provided with this paper.

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

## Acknowledgements

We would like to thank Leonhard Heinz (Medical University of Vienna, Austria), Sanjiv Luther (University of Lausanne) and all the members of the Rebsamen laboratory for critical discussions and suggestions, Giulio Superti-Furga (CeMM Research Centre for Molecular Medicine, Austria) for kindly providing critical reagents and suggestions, Pawel Pelczar and the Centre for Transgenic Models (University of Basel) for the generation of TASL and TASL2 knockout mice, Steven Porcelli (Albert Einstein College of Medicine, NY, USA) for providing Flt3L producing cell line, the Cellular Imaging Facility, the Flow Cytometry Facility, the Lausanne Genomic Technologies Facility, the Histology Facility and the CLE Animal Facility of the UNIL for technical support and the NIH Tetramer Core Facility (contract number 75N93020D00005) for providing LCMV specific H-2D(b) biotinylated monomers. Plasmids obtained through Addgene were a gift from D. Trono and F. Zhang. The work in M.R. laboratory was supported by the Swiss National Science Foundation (project grant 310030_200709) and the Fondation Pierre Mercier pour la science.

## Author contributions

Conceptualization: A.D., M.R. Methodology: A.D., L.B., M.R. Investigation: A.D., L.B., M.D., C. R-C., M.L., M.M.T., J.K., S.R. Software: S.C.C., J.M. Validation: A.D., L.B., M.D., M.L., M.M.T., J.K. Formal Analysis: A.D., S.C.C. Resources: J.M., M.R. Data curation: A.D., L.B., S.C.C., J.M., M.R. Writing – original draft: A.D., M.R. Writing – review & editing: A.D., L.B., M.D., S.C.C., M.M.T., J.M., M.R. Visualization: A.D., L.B., M.D., S.C.C., M.L., M.R. Supervision: J.M., M.R. Project administration: M.R. Funding acquisition: M.R.

## Competing interests

M.R. is an advisor of Solgate GmbH, Austria, and an inventor on European priority patent applications (EP 22 203 423.3, EP 22 203 422.5, EP 22 203 421.7, status: filed) covering small-molecule modulators of TASL and their medical use. The other authors declare no competing interests.
