## [Peer Review file · Nature Communications]

The TLR7/9 adaptors TASL and TASL2 mediate IRF5-dependent antiviral responses and autoimmunity in mouse

Corresponding Author: Professor Manuele Rebsamen

Version 0:

Reviewer comments:

Reviewer #1

(Remarks to the Author)

Various previous studies revealed that human TASL is an adaptor downstream of TLR7 and TLR9 that interacts with the endolysosomal transporter SLCL15A4, and is implicated in IRF5 activation, whereas in mice deletion of SLCL15A4 (feeble mice) abolishes responses of endosomal TLRs in pDCs and significantly reduces SLE disease in mouse models of lupus. In the current manuscript by Drobek et al., the authors aimed to investigate the pathophysiological role of murine TASL *in vivo*. To do so, they generated TASL-deficient (TASLko) mice and by comparing *ex vivo* the responses of TASLko pDCs and B cells to SLCL15A4-deficient (feeble) cells, upon TLR7 or TLR9 activation, they observed that in TASLko cells IRF5 activation was strongly reduced but not completely impaired as in SLCL15A4-deficient (feeble) cells, suggesting that in mice TASL redundancy might exist. Next, the authors identified a paralogue of murine TASL that they named it TASL2 and generated mice deficient for TASL2 (TASL2ko). By studying and comparing the responses of pDCs and B cells from TASL2ko, TASLko, double TASL/TASL2 (TASLDko) and feeble mice to TLR7 and TLR9 stimuli, they found that the responses of TASLDko cells mirrored the responses of feeble cells, TASLko cells also showed reduced responses but less profound compared to feeble or TASLDko counterparts, while TASL2ko cells showed almost a normal response. Furthermore, *in vivo* studies revealed that upon LCMV infection feeble and TASLDko mice were highly susceptible, while single TASL and TASL2ko deficiency were almost unaffected. In addition, pristane-induced lupus disease in four mouse genotypes revealed that the SLC15A4-TASL complex is implicated in disease development. The novelty of the study is based on exploring the signaling pathway of murine TASL and TASL2 that interact with SLC15A4 and their implication in LCMV infection and chemical-induced lupus disease. Overall, the data are of good quality and the manuscript is nicely written. However, the main message that there is redundancy between TASL and TASL2 upon LCMV infection, but not in the case of pristane-induced lupus is not fully supported by the current presented data, mainly because murine TASL and TASL2 are expressed at different levels in different cell types and that might have an effect on their importance depending on which cell type is studied. Thus, the following issues have to be addressed:

1. One of the main points of the study is that TASLko shows a more profound impact than TASL2ko in phenocopying SLC15A4 deficiency and IRF5 signaling. However, this conclusion is largely founded on studying two cell types -pDCs and B cells- in which TASL is highly expressed while the expression of TASL2 is minimal if any (Fig S3c). Thus, what are the responses of TASL2ko versus TASLko cells upon R848 and CpG stimulation in cells such as cDCs and splenic macrophages, where TASL2 is highly expressed while the expression of TASL is less important? It is essential to clarify if the degree of TASL and TASL2 contribution in cell activation upon TLR7 or TLR9 stimulation is cell type dependent or not?
2. In the same line, since TASL2 is highly expressed on cDCs and at very low levels in B cells, the characterization of signaling regarding IRF5 activation should be done primary on TASL2ko cDCs and secondary on B cells (Fig 3e, f, h, i). Studying IRF5 signaling in TASL2ko pDCs does not seem logical since TASL2 expression is absent in pDCs (Fig S3c).
3. Why TASLko and feeble cDCs show a significant reduced activation upon R848 stimulation (Fig. 5c) while previous studies established that feeble mutation selectively impairs the ability of pDCs, but not cDCs, to respond to TLR ligands (Blasius et al., PNAS, 2010, 107, 19973)?
4. The connection of production of IL-6 and TNF by TASLko B cells due to reduced number of B cells that produce these cytokines (Fig S2i) is confusing. Does the reduction in TASLko B cells is the outcome of a loss of a certain B cell subtype, or overall B cells produce less cytokines?

5. Why TNF and IL-6 production is reduced in SLC15A4 deficient B cells upon R848 stimulation, while the NF- κ B and MAPK pathways seem normal? Moreover, there is a discrepancy regarding IRF7 protein levels by SLC15A4 deficient pDCs upon R848 stimulation at 5h. IRF7 levels look normal in Fig 1a but reduced Fig 1c-d.
6. Anti-DNA and anti-RNA autoantibodies should be also evaluated in the sera of pristane-induced lupus disease mice, since the adaptors SLC15A4 and TASL act downstream of TLR9 that detects DNA and TLR7 that senses ssRNA.
7. Elevated levels of autoantibodies or the deposition of immune complexes do not necessarily indicate the presence of pathology. Histopathological evaluation of the kidneys of the pristane treated mice (TASLko, TASL2ko, TASLDko, feeble and WT) should be performed in order to validate the importance of SLC15A4 and TASL in SLE development. Moreover, it should be evaluated if the mice develop proteinuria.
8. Many essential data regarding the involvement of TASL2 in pristane-induced lupus disease have been omitted. Serum Sm/RNP autoantibodies and ANA antibodies, as well as immune complex deposition in the kidneys of TASL2ko mice should be included in Fig 7C, Fig7d-e and Fig 7f-g, as it has been done for the rest of the mice.
9. Based on the in vivo data in Fig S5, TASL2ko mice show a dramatic reduction ($\sim 1/4$ versus WT) of type I IFN levels in sera upon CpG injection, while upon R848 stimulation they show a normal response (Fig5a). Why this discrepancy between the two stimuli and the in vitro data where both stimuli seem to have similar a minimal effect on TASL2ko cells?
10. The age and the sex of the mice that were used for the conclusion that TASLko, TASL2ko and TASLDko mice do not develop any overt phenotype (Lines 123-125 and lines183-185) should be mentioned. Moreover, what is the immunological phenotype of older (6-12 months) TASLko, TASL2ko and TASLDko mice? Do they still look normal?
11. The term "TLR7-9" in the title and throughout the text is confusing because it is not clear if it refers to TLR7 and TLR9 or to TLR7, TLR8 and TLR9. Moreover, in the title of the paper it should be clarified that the studies refer to murine genes.
12. The sex and age of mice that used in the various experiments should be noted at the Figure legend of each figure, especially since SLE and LCMV infection show a sex bias and TASL gene is located in the X chromosome
13. Why BM-pDC from different deficient mice and assays are stimulated with different doses of R848? For the immunoblots in Fig 1a, SLC15A4 deficient pDCs are stimulated with 100ng/ml R848 for the immunoblots in Fig 2a, TASL deficient pDCs are stimulated with 5ug/ml R848 while for cytokine production with 100ng/ml R848.
14. Which study shows that "human system TASL deficiency fully-phenocopied SLC15A4 loss" (lines146-147)? Please add reference.
15. The experiments in Fig. 3c are based on overexpression of TASL2 in TASL-deficient cells, as such cannot be concluded that TASL2 can substitute TASL, unless TASL2 is expressed at normal levels. Overexpression studies to study the function of TASL2 protein is not the best approach since it can lead to interactions that do not happen under normal expression levels and thus wrong conclusions.
16. Clarify the two different wild-type groups (WT and WT control) that appear in Figure 5 and 6. Moreover, statistics are missing in Fig 2f and g.

Reviewer #2

(Remarks to the Author)

Drobek et al. provide a detailed analysis of the role of TASL, along with their newly described TASL2 (expressed in mice, not in humans), in immune system signaling and response in TASL, TASL2 and TASL/TASL2 double knock out mice after TLR7 and TLR9 activation. They particularly focus on the role of TASL in IRF5 activation and cytokine and IgG2 production in vitro and in vivo. They also study the role of TASL and TASL2 in vivo after LCMV infection or pristane injection to investigate those components of endosomal TLR signaling pathways in relevant virus infections and in a model of lupus-like disease.

The experiments are nicely performed with data generally compared to data from wild type mice and the "feeble" SLC15A4-deficient mouse. Overall, the data implicate the SLC15A4/TASL complex in IRF5 activation, but not NF κ B activation, with comparable results presented for the feeble mouse and the TASL/TASL2 double knockout mouse. Data are more variable for TASL vs. TASL2. Although TASL appears to have a dominant effect compared to TASL2 for most experimental outcomes, there are differences noted (and not always emphasized in the text) with regard to cell distribution of the two gene products (Fig. S3c and d) and response to in vivo administration of TLR7 (848) vs. TLR9 (CpG) ligands (Figs. 5 and S5). As it appears that there are some differences in immune mediators induced depending on whether signals are provided by TLR7 vs. TLR9 ligands, it would be helpful if the authors could critically describe any differences revealed with regard to those two pathways and the relative contributions of TASL vs. TASL2 in cytokine and IgG induction.

The two murine systems chosen for study of host defense and autoimmunity, LCMV infection and pristane-induced lupus,

are both relevant to TLR7 stimulation. For that reason the models are informative for investigation of TASL. But it is important that the authors strongly make the point that infection with other viruses or other murine lupus models might show very different requirements for TASL and IRF5 in immune activation and tissue pathology.

With regard to the pristane lupus data shown in Fig. 7, the images for ANA (panel d) and immunoglobulin deposition in the kidney (panel f) are inadequate. Other than the control pristane data the images appear totally black, which does not provide confidence that the cell staining for ANA and tissue sections and staining for the IgG deposition in the kidney are appropriately prepared. In any case, if the authors intend to convincingly show the dramatic abrogation of autoimmune disease pathology that they suggest, they should also provide data on proteinuria, the typical measure of renal pathology in murine lupus studies. In addition, while the data for anti-Sm/RNP are dramatic, suggesting a strong dependence on TASL and effective TLR7 signaling for production of that specificity (for an RNA-binding protein antigen), it would be helpful to also specifically measure anti-dsDNA autoantibodies in the different experimental conditions. The contribution of TLR7 vs. TLR9 to development of clinical lupus disease based on signaling through each of those endosomal TLR signaling pathways and on development of RNA vs. DNA autoantibody specificities is of great interest in understanding mechanisms of development of autoimmunity in SLE. In some models TLR9 provides protection from autoimmunity while TLR7 drives both type I interferon and autoantibody production. As the authors have established systems to separately use TLR7 and TLR9 stimuli it will be important to elaborate on any distinctions noted in cytokine and IgG production after administering TLR7 vs. TLR9 stimuli (the data in Figs. 5 and S5) and to provide data on RNA-targeted (anti-Sm/RNP) vs. DNA-targeted (anti-dsDNA) autoantibodies in the pristane model experiments.

Reviewer #3

(Remarks to the Author)

In this paper, the authors explore the SLC15A4-TASL axis and its effects on TLR7-9 induced IRF5 activation following deficiency or knockout. One of the biggest findings of this study is the characterization of TASL and its paralogue TASL2 (Gm6377), demonstrating their distinct contributions to TLR7-9 signaling pathways. The experimental findings, particularly those regarding TASL and TASL2 knockout mouse models, provide compelling evidence for the critical roles of these molecules in modulating immune responses. The observed phenotypic similarities between TASL-deficient, TASL2-deficient, and SLC15A4-deficient mice underscore the importance of TASL-mediated IRF5 activation in regulating TLR7-9-driven inflammatory cascades. Overall, this research provides ground-breaking contributions to the field of immunology, elucidating the intricate mechanisms underlying TLR7-9 signaling and its implications in anti-viral immune clearance as well as for autoimmune pathogenesis.

Comments:

- 1) Unclear to the reviewer why authors used R848 at a concentration of 10 $\mu\text{g/ml}$ in Fig 3c-d, where elsewhere was used at 100 ng/ml.
- 2) Fig1d is missing isotype controls in both mice. Also, while the feeble mice flow staining was lower with IRF7, a significant shift in IRF7 was still observed post R848 treatment in 1d. The authors need to discuss this point and compare the shifts in the signals between times 0 and 5h. This is not clear especially as the authors argue that IRF7 protein induction was diminished in (Fig. 1c-d). Related to this, in western blot following R848 there was no visible difference shown for IRF7 which need to be clarified by the authors. In figure 1c, the authors do not comment on how/why IRF7 levels are similar between wildtype and feeble mice @ 24h of R848 stimulation.
- 3) Fig1s, information to replicate this figure for the reader in S1a-b are lacking.. how long the B cells were incubated in the presence of Brefeldin A (conc'n is missing), how long after r848/ CPG treatment, was BFA added? Why are only 20% of the cells responding to the R848 WRT IL-6 while 10% were positive TNF. Not to mention the very low MFI (100?) was an isotype control used?.. can you show the raw data?
- 4) In fig d-e, Splenic pDC were activated with R848 in the presence of Brefeldin A, but the authors seem to pick and choose what result to show in D it is % and E is MFI. It will be more consistent to show both for both cytokines and discuss why only 5-10 % are responding to the R848 treatment for example.
- 5) LCMV cytotoxic T cell responses is key in LCMV clearance, yet the authors could not detect differences as measured by LCMV-specific tetramers which suggests other functional assays would provide better answers. This should be discussed.

Version 1:

Reviewer comments:

Reviewer #1

(Remarks to the Author)

In the revised version of the manuscript the authors addressed many of my comments, but still show a strong bias on emphasizing the role of TASL, while trying to minimize the role of TASL2, both while presenting the data and during the discussion. I agree with the authors that TASL/TASL2 double deficiency phenocopies SLC15A4 deficiency, but it is also important to clearly present in an unbiased manner the independent contribution of TASL and TASL2 in the various assays and cell types. The authors are the first ones that identified TASL2 and generated deficient mice and so they are in the great

position to clarify the independent function of TASL and TASL2. Moreover, based on the fact that in the various studies both sexes of mice were mixed is raising an important issue regarding the validity of the data.

Points that still have to be addressed:

1. The new data that were generated by answering my previous Points 1-4 regarding the differential redundancy between TASL and TASL2 in the various cell types including cDCs, macrophages and B cells should be appropriately presented and discussed in the manuscript and not only in the point-by-point response to the reviewer.
2. Previous points 6 and 7 have not been fully addressed. As the authors mention pristine-lupus model is not ideal in the C57BL/6 background, thus at least maximum possible data are needed to support correct conclusions. In their rebuttal letter the authors responded "We would like to point out that the conclusion that TASLKO has more profound impact than TASL2KO is also very strongly supported by the in vivo Results", however, no in vivo data are provided for the TL2KO mice for certain experiments. The contribution of TASL2 in lupus development should be clarified by presenting sera autoantibodies, ANA antibodies and kidney histology from TL2ko mice, as has been requested in the revision of the original manuscript and as it has been done for TASL deficient mice (Figure 7 c-g). Moreover, histopathological scoring for glomerulonephritis in the kidneys of all mouse groups has to be done by a pathologist. The data provided in Figure 3 in the rebuttal letter are not convincing, magnification is too low and scoring not detailed. The argument of the authors that previous studies did not reveal glomerulonephritis in SLC15A4 deficient mice is not valid since microbiota plays an important role in autoimmune development and the microbiota vary dramatically from one animal facility to another.
3. Previous Point 11 has not been addressed. Human and mouse genes and proteins are not the same and since the studies performed in mice this should be mentioned in the title of the paper.
4. Previous point 12 regarding sex and age of mice used in the studies and the request that this information have to be added in all the Figure legends. In the revised versions and in many figure legends now it is written: "Both males and females were used as a source of primary cells". Mixing genotypes for the various assays is not appropriate since it is not obvious at what degree the differences that are observed between the genotypes are the outcome of the genotype and/or the sex, since immune responses vary depending on the sex. More specific, it is known that pDCs from female mice and humans have higher basal levels of IRF5 and IFN-alpha production following TLR7 stimulation (Griesbeck et al., J Immunol, 2015, 195:5327). Thus, using only females or more females in one genotype than in the other can give a bias for higher TLR7 signaling for this genotype. Moreover, even for the in vivo studies - LCMV infection and pristane -induced lupus - the authors used both sexes. Mixing both sexes it is also not appropriate for the current study since TLR7 is located in the X chromosome both in mice and humans. It is well established that the TLR7 gene, encoded in the X chromosome both in mice and humans, can escape X inactivation resulting in higher expression of TLR7 in females than males that can partially explain the higher incidence of lupus in females (Pisitkun et al., Science, 2006, 312:1669). Thus, the mice used for each experiment have to be of the same sex for all the genotypes.

Reviewer #2

(Remarks to the Author)

The authors have done an excellent job of responding to all reviewer comments. They have performed extensive additional experiments and have very thoughtfully discussed the comments and questions raised by the reviewers. It will be important for future studies to investigate TASL and TASL2 in additional murine models of lupus as well as additional models of virus infection.

Reviewer #3

(Remarks to the Author)

Version 2:

Reviewer comments:

Reviewer #1

(Remarks to the Author)

Data on old TL2ko mice regarding lupus development (sera autoantibodies, ANA antibodies and kidney histology) still have not been provided. The reply of the authors very clearly explains why, but still does not resolve the issues that I raised in my previous reports.

Moreover, the issue on performing experiments by mixing cells of mice from both sexes remains. Many experiments were performed by mixing female and male cells/mice despite the fact that TLR7 is located on the X chromosome. It is known that female cells show higher response than males upon TLR7 stimulation and in certain degree the increased TLR7 signaling in females is responsible for the higher incidence of lupus in females than males. Based on the argumentation of the authors

in their rebuttal letter, I agree that for a phenotype that is profound like in the case of TASLDKO and feeble versus wild type mice mixing both sexes maybe is not so vital. However, for a modest phenotype like in the case of TLSL2ko mice this can have detrimental consequence on driving correct conclusions.

We thank the Reviewers for the careful and constructive assessment of our study. We are grateful for the insightful comments, which we have addressed in the detailed point-by-point reply here below and in the revised manuscript. We believe that the suggested experiments strongly contributed to further improve our study.

Reviewer 1:

*Various previous studies revealed that human TASL is an adaptor downstream of TLR7 and TLR9 that interacts with the endolysosomal transporter SLCL15A4, and is implicated in IRF5 activation, whereas in mice deletion of SLCL15A4 (feeble mice) abolishes responses of endosomal TLRs in pDCs and significantly reduces SLE disease in mouse models of lupus. In the current manuscript by Drobek *et al.*, the authors aimed to investigate the pathophysiological role of murine TASL *in vivo*. To do so, they generated TASL-deficient (TASLko) mice and by comparing *ex vivo* the responses of TASLko pDCs and B cells to SLCL15A4-deficient (feeble) cells, upon TLR7 or TLR9 activation, they observed that in TASLko cells IRF5 activation was strongly reduced but not completely impaired as in SLCL15A4-deficient (feeble) cells, suggesting that in mice TASL redundancy might exist. Next, the authors identified a paralogue of murine TASL that they named it TASL2 and generated mice deficient for TASL2 (TASL2ko). By studying and comparing the responses of pDCs and B cells from TASL2ko, TASLko, double TASL/TASL2 (TASLDko) and feeble mice to TLR7 and TLR9 stimuli, they found that the responses of TASLDko cells mirrored the responses of feeble cells, TASLko cells also showed reduced responses but less profound compared to feeble or TASLDko counterparts, while TASL2ko cells showed almost a normal response. Furthermore, *in vivo* studies revealed that upon LCMV infection feeble and TASLDko mice were highly susceptible, while single TASL and TASL2ko deficiency were almost unaffected. In addition, pristane-induced lupus disease in the four mouse genotypes revealed that the SLC15A4-TASL complex is implicated in disease development. The novelty of the study is based on exploring the signaling pathway of murine TASL and TASL2 that interact with SLC15A4 and their implication in LCMV infection and chemical-induced lupus disease. Overall, the data are of good quality and the manuscript is nicely written. However, the main message that there is redundancy between TASL and TASL2 upon LCMV infection, but not in the case of pristane-induced lupus is not fully supported by the current presented data, mainly because murine TASL and TASL2 are expressed at different levels in different cell types and that might have an effect on their importance depending on which cell type is studied.*

*We thank the Reviewer for the comments and for acknowledging the novelty and the quality of the data presented. We would like to stress that one of the main messages of our work is that TASL/TASL2 double deficiency fully phenocopied SLC15A4 deficiency in all the investigated settings. Indeed, while the importance of TASL is supported by our previous *in vitro* work in the human system (Heinz *et al.*, Nature 2020 PMID: 32433612; Zhang *et al.*, Cell Reports 2023 PMID: 37527038; Boeszoermenyi *et al.*, Nat Commun 2023 PMID: 37863876), its central role for SLC15A4-mediated responses is still a debated question, as other laboratories have proposed alternative mechanisms independent from TASL function, as summarized in recent reviews published after our identification of TASL in 2020 (Kobayashi *et al.*, Front Immunol 2023, PMID: 37781390; Toyama-Sorimachi, Nat Chem Biol. 2024, PMID: 38347215; Toyama-Sorimachi, Int Immunol. 2024, PMID: 38946351).*

Concerning the redundancy between TASL and TASL2, we fully agree that the different expression levels of TASL and TASL2 in different cell types is of interest and we extensively addressed this point in the revised version of the manuscript. For this, we investigated cell responses in individual cell types as requested by the Reviewer, providing new data on cDCs and macrophages as detailed in the point-by-point responses here below. Regarding the differential redundancy of TASL and TASL2 in the LCMV and pristane models mentioned by the Reviewer, we would like to point out that this is strongly supported by the clarity of the phenotypes observed in TASL^{KO}, TASL2^{KO} and TASL^{DKO} in these two models. Indeed, in the LCMV model, single TASL and TASL2 knockouts are able control infection while TASL^{DKO}, like *feeble*, are not, strongly supporting a redundant/compensatory role for these two TASL paralogues. In contrast, in the pristane-induced SLE model, single TASL deficiency is sufficient to block all pathogenic manifestations (monocyte infiltration, ISG signature, autoantibody formation,...) to a level comparable to *feeble* and TASL^{DKO}, indicating that TASL2 is not sufficient to compensate for TASL loss in these settings. As mentioned in the discussion (lines 406-414), we hypothesize that in the pristane model, even a partial reduction in IRF5 activation (as we observed in TASL^{KO}) is sufficient to prevent disease manifestation, as supported by previous observations that haploinsufficiency of IRF5 or SLC15A4 is similarly protective (Richez *et al*, J Immunol 2009, PMID: 20007534; Pellerin *et al*, JCI Insight 2021, PMID: 34197340; Katewa *et al*, PLoS One, PMID: 33444326). As mentioned above, in our opinion the key finding of our study is that TASL double deficiency (which mimics TASL^{KO} in humans) fully phenocopied SLC15A4 *in vivo*, demonstrating therefore the essential role of TASL-SLC15A4 complex for TLR7-9 responses and clarifying the elusive function of SLC15A4 in this pathway. This information is of high relevance considering current efforts to target SLC15A4 in context of human lupus and related diseases (Boeszoermyeni *et al*, Nat Commun 2023, PMID: 37863876, Chiu *et al*, Nat Chem Biol 2024, PMID: 38191941; Wang *et al*, J Med Chem 2020, PMID: 31774679; Toyama-Sorimachi Nat Chem Biol. 2024, PMID: 38347215; Toyama-Sorimachi Int Immunol. 2024, PMID: 38946351).

Thus, the following issues have to be addressed:

1. One of the main points of the study is that TASLko shows a more profound impact than TASL2ko in phenocopying SLC15A4 deficiency and IRF5 signaling. However, this conclusion is largely founded on studying two cell types -pDCs and B cells- in which TASL is highly expressed while the expression of TASL2 is minimal if any (Fig S3c). Thus, what are the responses of TASL2ko versus TASLko cells upon R848 and CpG stimulation in cells such as cDCs and splenic macrophages, where TASL2 is highly expressed while the expression of TASL is less important? It is essential to clarify if the degree of TASL and TASL2 contribution in cell activation upon TLR7 or TLR9 stimulation is cell type dependent or not?

We thank the Reviewer for this observation. We would like to point out that the conclusion that TASL^{KO} has more profound impact than TASL2^{KO} is also very strongly supported by the *in vivo* results. Indeed, TASL deficiency had much stronger impact than TASL2^{KO} on all responses assessed upon *in vivo* TLR7 and TLR9 stimulation (Fig. 5 and S6). We initially focused our *ex vivo* work on pDC and B cells because these are the cell types that, on one side, are among the main contributors to SLE, playing a central role in production of Type I IFN and autoantibodies, respectively (Vinuesa *et al*, Science 2023, PMID: 37141353) and, on the other, were the main focus of previous reports using *feeble*/SLC15A4-deficient mice (Blasius *et al*, PNAS 2010, PMID: 21045126; Baccala *et al*, PNAS 2013, PMID: 23382217, Kobayashi *et al*, Immunity 2014, PMID: 25238095).

At the same time, we agree that it is of interest to also investigate the role of TASL and TASL2 in cDCs and splenic macrophages. We have included new data on these cell types in the revised version of the manuscript (see new figure S5a-e and S6d-e).

Concerning the cDCs, we now provide data on:

- Splenic cDCs upon *ex vivo* TLR stimulation:
 - o TLR7 and TLR9-induced cytokine production is blunted in $TASL^{DKO}$ and *feeble*, while single deficiency of *TASL* or *TASL2* has only minor effect, supporting redundant role (Fig. S5a). Of note, while *TASL2* is expressed at higher level in cDCs, we would like to point out that *TASL* is also present in these cells (data in Fig. S3d shows its relative expression to a highly expressed gene, *HPRT*).
- Bone-marrow-derived cDCs (Flt3L-differentiated, MACS Siglec-H+ depleted) upon *ex vivo* TLR stimulation (Fig. S5b-d):
 - o TLR7 and TLR9-induced cytokine production: $TASL^{DKO}$ phenocopied the reduction observed in *feeble* cells, while single *TASL* KO show a partial impairment (Fig. S5b).
 - o TLR7 and TLR9-induced IRF5 activation is blunted in $TASL^{DKO}$ and *feeble*, while partially affected in single *TASL* knockout cells (Fig. S5c-d). Of note, R848 stimulation in these cells consistently resulted in weak activation of IRF5 when compared to other cell type tested (BM-pDC, B cells). Nevertheless, the impairment in $TASL^{DKO}$ was consistently observed (Fig. S5b-d).
- ISG induction in splenic cDCs upon *in vivo* TLR7/9 stimulation: To further investigate cDCs, we assessed the induction of ISG upon *in vivo* TLR stimulation in splenic cDCs (as previously done in splenic pDCs, Fig. 5b and S6c). As shown in new Fig. S6d-e, the ISG signature 24 hours after i.v. injection of R848 or CpG-B was blunted in $TASL^{DKO}$ and *feeble* cDCs, very strongly affected in single $TASL^{KO}$ and largely normal in $TASL2^{KO}$. This is largely consistent with the serum level of Type I IFN we measured (Fig. 5a and Fig. S6a), which are primarily secreted by pDCs (Asselin-Paturel and Trinchieri, J Exp Med 2005, PMID: 16103406). Therefore, this strongly suggests that the observed ISG induction and upregulation of activation marker in splenic cDC in these settings is mostly an effect of paracrine signaling from circulating IFN/cytokines rather than cDC intrinsic TLR stimulation.

Lastly, we investigated splenic macrophages and showed that they express *Tas1* and, at a higher level, *Tas2* (Fig. S3d). TLR7 and TLR9-induced cytokine production in splenic macrophages stimulated *ex vivo* showed blunted responses in $TASL^{DKO}$ and *feeble*, and almost comparable effect in $TASL^{KO}$, while the impact of $TASL2^{KO}$ appeared to be more cytokine dependent (reduction of IL-12p40 and TNF, while no effect on IL-6) (Fig. S5e).

The different impact of *TASL* and *TASL2* and cell type-dependent response has been discussed in the revised manuscript (lines 362-375 and 408-414).

2. In the same line, since *TASL2* is highly expressed on cDCs and at very low levels in B cells, the characterization of signaling regarding IRF5 activation should be done primary on *TASL2ko* cDCs and secondary on B cells (Fig 3e, f, h, i). Studying IRF5 signaling in *TASL2ko* pDCs does not seem logical since *TASL2* expression is absent in pDCs (Fig S3c).

As described above, we have now comprehensively assessed the effect of *TASL* and *TASL2* in cDCs, including the activation of IRF5 (Fig. S5c-d).

Concerning the rationale of assessing the role of *TASL2* in B cells and pDCs, this arises from the fact that we observed residual IRF5 activation in $TASL^{KO}$, as shown in B cells (Fig. 3e-f) as well as in BM-pDCs (Fig. 3h-i), while the impairment was complete in *feeble*. Indeed, $TASL^{DKO}$ was required for a complete block of IRF5 activation and phenocopied *SLC15A4* deficiency, therefore confirming a partial role for *TASL2* in these two cell types (Fig. 3e-i). Importantly, this is fully supported by the transcriptional profile shown in Fig. 4 and Fig. S4. Of note, we would like to clarify

that TASL2 is not absent in splenic pDCs, only its expression is comparatively lower than in cDCs, as we confirmed by our own qPCR data (Fig. S3d, which shows relative expression to the highly expressed reference gene HPRT).

We have now further investigated the expression of the two paralogues in B cells by directly quantifying them by qPCR, which confirmed that TASL2 is also expressed (Fig. S3d - please also note that in Fig. S3c showing the expression data from the Immgen database the scale for TASL and TASL2 is different).

Therefore, the expression of both *Tasl* and *Tasl2* in B cells and pDCs is fully consistent with the impact on TLR7/9 responses observed in the different KOs.

3. Why TASLko and feeble cDCs show a significant reduced activation upon R848 stimulation (Fig. 5c) while previous studies established that feeble mutation selectively impairs the ability of pDCs, but not cDCs, to respond to TLR ligands (Blasius et al., PNAS, 2010, 107, 19973)?

We would like to clarify that the experimental settings between our experiment shown in Fig. 5c and the work from Blasius and colleagues (PNAS 2010, PMID: 21045126) are different. Indeed, while Blasius *et al.* assessed TNF production in GM-CSF bone-marrow derived cDCs treated *in vitro*, our data showed splenic cDC activation (monitored by CD86 upregulation) after *in vivo* stimulation by i.v. injection of R848. As mentioned above, the cDCs activation and CD86 upregulation we observed upon stimulation *in vivo* is most likely indirect and the result of the strong cytokine production by pDCs and other cells observed in WT mice, which is impaired in *feeble*, TASL^{KO} and TASL^{DKO} (similarly to the bystander activation observed in T cells, Fig. S6h).

Concerning the relevance of SLC15A4 for TLR7 and TLR9 responses in cDCs, we would like to mention that this has been controversial, with different studies investigating SLC15A4-deficient cells reporting contrasting results. As mentioned by the Reviewer, Blasius *et al.* (PNAS 2010, PMID: 21045126) did not observe a major effect on TNF production by GM-CSF bone-marrow derived cDCs from *feeble* mice stimulated *in vitro*. In contrast, another study (Sasawatari *et al.* Gastroenterology 2011, PMID: 21277849) used bulk Flt3L-derived DC, showing clear defect in several cytokines (IL-12p40, TNF) as well as *Ifnb*. Furthermore, Dosenovic *et al.* investigated primary splenic cDCs and observed a strong reduction in TNF and IL12p40 levels in *feeble* cDCs upon *in vitro* TLR9 stimulation of splenocytes by CpG-A/B (Immunol Cell Biol. 2015, PMID: 25310967). Lastly, Katewa *et al.* (PLOS One 2021, PMID: 33444326) also reported a partial reduction (50%) in cytokine production (IL6, IL12p40, TNF, CCL2) in myeloid DCs (GM-CSF derived) upon R848 stimulation.

As detailed above, our new data shows a strong impairment of cytokine (IL-6, IL12p40 and TNF) production in FACS-purified splenic cDC from *feeble* and TASL^{DKO} stimulated *in vitro* with R848 or CpG-B (Fig. S5a). Significant reduction was also observed in Flt3L-induced BM-cDC enriched by MACS (Siglec-H+ cells depletion) (Fig. S5b). Therefore, our data show that the SLC15A4-TASL/TASL2 pathway is crucial for IRF5 activation in cDC.

4. The connection of production of IL-6 and TNF by TASLko B cells due to reduced number of B cells that produce these cytokines (Fig S2i) is confusing. Does the reduction in TASLko B cells is the outcome of a loss of a certain B cell subtype, or overall B cells produce less cytokines?

Concerning the B cell subtypes, we did not observe any reduction in splenic subpopulations (IgM vs IgD) in TASL deficient mice at steady state (see here below Reviewer 1 – Figure 1). Moreover, we have now confirmed that the reduction in IL-6 production is observed in both IgM+ and IgD+

subpopulations, indicating therefore general reduction and not subtype specific (Reviewer 1 – Figure 2).

Lastly, we would like to mention that the results of the intracellular staining in TASL^{KO} B cells (shown in original Fig. S2i, now Fig. S2j) are largely consistent with the IL6 ELISA data (original Fig. S3k, now Fig. S3l) as well as with the results obtained in *febl/e* B cells (Fig 1e, and original Fig. S1a-b, now Fig. S1e-f). Moreover, these findings are in line with data by Kobayashi *et al.* (Immunity 2014, PMID: 25238095) and Katewa *et al.* (PlosONE 2021, PMID: 33444326) showing reduced IL6 and TNF levels in TLR7- and/or TLR9-stimulated SLC15A4-deficient B cells.

Reviewer 1 – Figure 1: Example of steady state IgM+ and IgD+ distribution in WT and TASL^{KO} splenic B cells.

Reviewer 1 – Figure 2: **a**, Example of IgM+ and IgD+ distribution in WT and TASL^{KO} splenic B cells treated *in vitro* with R848 O/N. **b**, Gating strategy for IL-6+ cells in total splenic B cells. **c-d**, IL-6+ cells in IgM+ and IgD+ subpopulations showing decrease of IL-6 producing cells in both TASL^{KO} B cell subtypes upon O/N stimulation with R848 (c) or CpG (d).

5. Why TNF and IL-6 production is reduced in SLC15A4 deficient B cells upon R848 stimulation, while the NF- κ B and MAPK pathways seem normal? Moreover, there is a discrepancy regarding IRF7 protein levels by SLC15A4 deficient pDCs upon R848 stimulation at 5h. IRF7 levels look normal in Fig 1a but reduced Fig 1c-d.

Concerning the TNF and IL-6 production, our data are fully consistent with previous reports by Kobayashi *et al.*, (Immunity 2014, PMID: 25238095) and Katewa *et al.* (PlosONE 2021 PMID:

33444326) who similarly reported normal NF- κ B activation, while reduced IL6 and TNF induction in SLC15A4-deficient B cells. The contribution of IRF5 activation to TNF and IL6 production in B cells is in line with what we and others observed in other SLC15A4, TASL or IRF5-deficient cell types (pDCs,...), which also show normal NF- κ B and MAPK activation (Katewa *et al.*, PlosONE 2021 PMID: 33444326). Indeed, there are evidence that IRF5 can cooperate with RelA to regulate a subset of inflammatory genes and that IRF5 can bind to cell type-specific enhancers (Saliba *et al.*, Cell Rep 2014, PMID: 25159141; Khoyratty & Udalova, Int. J. Biochem. Cell Biol. 2018, PMID: 29578052). This was for example shown to contribute to TNF regulation in human dendritic cells (Krausgruber *et al.* Blood 2010, PMID: 20237317).

Concerning IRF7 level, we have further confirmed that induction of IRF7 upon stimulation with R848 or CpG-B is indeed reduced in *feeble* BM-pDC. We further analyzed IRF7 level by WB including additional timepoints and could confirm the partial, but reproducible reduction in IRF7 induction upon R848 and CpG-B treatment (Fig. S1a-b), supporting the original data in Fig. 1a-b and 1c-d, now Fig. S1c-d. Moreover, to support our data on intracellular IRF7 staining, we now included isotype control staining (Fig. 1c-d). Of note, our transcriptional profile data confirm the reduction of IRF7 induction in *feeble* BM-pDC (Fig. 4e and S4f). When considered altogether, we think that these data consistently indicate that SLC15A4 deficiency results in partial reduction of IRF7. The residual induction is most likely due to, on one side, the extreme sensitivity of IRF7 transcriptional regulation to interferons and, on the other, the fact that the *feeble* cells IFNAR signaling is reduced but not absent as shown by phospho-STAT1 (Fig. 1a-b).

6. Anti-DNA and anti-RNA autoantibodies should be also evaluated in the sera of pristane-induced lupus disease mice, since the adaptors SLC15A4 and TASL act downstream of TLR9 that detects DNA and TLR7 that senses ssRNA.

We agree with the reviewer that this is an important point and we included these new data in the revised version of the manuscript (new Fig. 7c). Supporting the critical role for TASL, we observed a strong reduction for both anti-DNA and anti-RNA antibodies, as well as in autoantibodies anti-Su (Ago2), a cytoplasmic antigen also observed in SLE patients (Satoh *et al.*, Adv Exp Med Biol 2013, PMID: 23224964).

7. Elevated levels of autoantibodies or the deposition of immune complexes do not necessarily indicate the presence of pathology. Histopathological evaluation of the kidneys of the pristane treated mice (TASLko, TASL2ko, TASLDKO, feeble and WT) should be performed in order to validate the importance of SLC15A4 and TASL in SLE development. Moreover, it should be evaluated if the mice develop proteinuria.

We thank Reviewer for this comment as indeed autoantibody levels and immunoglobulin depositions are key drivers of disease and precede development of kidney pathology which depends on murine model used and the mouse strain. We have edited manuscript accordingly to be more precise when discussing this point (line 314-327). Moreover, we would like to mention that the importance of SLC15A4 in pristane induced SLE models have been previously established (Kobayashi *et al.*, Immunity 2014; Katewa *et al.* PlosONE 2021). For this reason, we decided to use the pristane-driven model also to assess the impact of TASL deficiency. In the C57Bl/6J background, pristane challenge allows to monitor many relevant features comprising the production of interferon by inflammatory monocytes, the generation of autoantibodies and the deposition of immunocomplexes in kidney. All these readouts were indeed affected by TASL deficiency. It is well reported in the literature that the C57Bl/6J strain is relatively resistant to kidney

pathology compared to other strains (like Balb/c) and that pristane, while inducing typical inflammatory and autoimmune SLE features, results in no or minimal kidney damage even in WT mice (Reeves *et al.* Trends Immunol 2009, PMID: 19699150). As expected, also in our experiments, pristane injected WT mice did not develop detectable proteinuria (therefore preventing assessment of the protective role of the knockouts). In line with this, histopathology evaluation of kidney (H&E staining, inflammatory infiltrate) failed to detect any significant increase between pristane injected WT and control, PBS injected littermates (nor across the knockouts) (see Reviewer 1 - Figure 3). Again, this is expected and fully consistent with the absence of overt glomerulonephritis in pristane treated C57Bl/6J reported for example by Katewa *et al.* (PlosONE 2021, PMID: 33444326) when investigating SLC15A4 deficient mice. As described here above (Reviewer 1, point 6), the demonstration of impaired generation of anti-RNA, anti-dsDNA and anti-Su antibody (new Fig. 7c) further support the key role of TASL (and SLC15A4) in pristane induced SLE model.

Reviewer 1 – Figure 3: **a**, Evaluation of inflammation score in kidney sections stained with Hematoxylin & Eosin (H&E). **b**, Example of scoring in H&E stained kidney sections.

8. Many essential data regarding the involvement of TASL2 in pristane-induced lupus disease have been omitted. Serum Sm/RNP autoantibodies and ANA antibodies, as well as immune complex deposition in the kidneys of TASL2ko mice should be included in Fig 7C, Fig7d-e and Fig 7f-g, as it has been done for the rest of the mice.

The effect of TASL2 deficiency has been fully investigated in the short-term analysis (2 weeks, Fig 7a-b and Fig, S8a-b). This did not indicate any effect of TASL2 loss on the different readouts, as TASL2^{KO} showed identical responses as WT animals. In contrast, single TASL deficiency fully phenocopied *feeble*, showing impaired responses. Moreover, the long-term analysis (6 months) revealed that single TASL^{KO} mice similarly phenocopied SLC15A4 deficiency (as well as TASL^{DKO}) and conferred full protection in all readouts. These results demonstrate that the TASL-dependent pathway is required for disease development *in vivo* (similarly as SLC15A4 deficiency), and therefore strongly support the notion that SLC15A4-TASL complex is a relevant therapeutic target for the human disease, which was the main goal of these experiments. Given the absence of phenotype in TASL2^{KO} in short term analysis and the overall duration of the long-term experiment (preparation of experimental cohorts + 2 months of age + 6 months treatment) we believe that these data are not essential to support the conclusion of this first manuscript, which show that the SLC15A4-TASL pathway is central in this chemical induced SLE model.

9. Based on the *in vivo* data in Fig S5, TASL2ko mice show a dramatic reduction (~ 1/4 versus WT) of type I IFN levels in sera upon CpG injection, while upon R848 stimulation they show a normal response (Fig5a). Why this discrepancy between the two stimuli and the *in vitro* data where both stimuli seem to have similar a minimal effect on TASL2ko cells?

We thank the reviewer for pointing this out. After careful evaluation of the original data, we detected the presence of an outlier with very high levels of type I IFN in the WT group, which contributed to exacerbate the difference and relative reduction observed in TASL2^{KO} group. For this reason, we increased the number of experimental data. After including additional mice, the difference was still present but less pronounced reaching statistical significance only for IFN β (now in Fig. S6a). Here below are presented the results including the original outlier (as presented in revised manuscript Fig. S6a) and without for Reviewer's assessment. To further investigate this point, we have extended our investigation to ISG induction in sorted splenic cDC (Fig. S6d-e). Overall, interferon signaling in TASL2^{KO}, albeit slightly reduced, was still consistently higher than in TASL^{KO} and TASL^{DKO} (Fig. S6c, e). Therefore, we think that, collectively, these data do not show major discrepancy between *in vivo* and *in vitro* data. Nevertheless, as pointed out by the Reviewer, we observed across different experiments a trend suggesting stronger impact of TASL2 deficiency upon CpG-B treatment compared to R848. While we believe that deeper investigation of this potential specific effect on TLR7 vs TLR9 signaling goes beyond the scope of this study, we have now mentioned this point in revised discussion of manuscript as one of the interesting questions to be addressed in follow-up studies (line 373-375).

Reviewer 1 – Figure 3: **a**, Serum levels of indicated IFNs, chemokines and cytokines 4h after CpG-B i.v. administration. Data are as they appear in revised manuscript Fig. S6a. **b**, Data as in (a) without the outlier in all 8 graphs.

10. *The age and the sex of the mice that were used for the conclusion that TASLko, TASL2ko and TASLDko mice do not develop any overt phenotype (Lines 123-125 and lines183-185) should be mentioned. Moreover, what is the immunological phenotype of older (6-12 months) TASLko, TASL2ko and TASLDko mice? Do they still look normal?*

The age and the sex of adult mice used for the immunophenotyping experiment presented in Fig. S2d, Fig. S3i and j has been added in figure legend. Moreover, we performed the same analysis on older mice (6-8 months) as suggested by the Reviewer and we did not observe any major effect (new Fig. S3k).

11. *The term “TLR7-9” in the title and throughout the text is confusing because it is not clear if it refers to TLR7 and TLR9 or to TLR7, TLR8 and TLR9. Moreover, in the title of the paper it should be clarified that the studies refer to murine genes.*

We agree with the Reviewer that the term “TLR7-9” was unclear and we clarified this in the revised version of the manuscript. We now use “TLR7-9” to indicate “TLR7, TLR8 and TLR9” in the human context and “TLR7/9” to indicate “TLR7 and TLR9” when referring to the murine situation. Concerning the title, we would like to refer to the proteins and not the genes, therefore we used capitalized form as official nomenclature and not the italic form used for murine genes.

12. *The sex and age of mice that used in the various experiments should be noted at the Figure legend of each figure, especially since SLE and LCMV infection show a sex bias and TASL gene is located in the X chromosome.*

We agree that this is an important missing information. We have now included statement specifying sex of mice in Figure legends. General range of the age of mice used in experiments is stated in Methods section.

13. *Why BM-pDC from different deficient mice and assays are stimulated with different doses of R848? For the immunoblots in Fig 1a, SLC15A4 deficient pDCs are stimulated with 100ng/ml R848 for the immunoblots in Fig 2a, TASL deficient pDCs are stimulated with 5ug/ml R848 while for cytokine production with 100ng/ml R848.*

We have now included immunoblots of TASL-deficient pDCs stimulated with 100 ng/ml R848 (new Fig. 2a). The results are fully consistent with the initial results obtained with 5µg/ml R848 stimulation (original Fig 2a, now Fig S2e). These initial experiments in TASL^{KO} were performed with high doses of R848 to induce maximal TLR stimulation and confirm that the strong impairment observed was dose independent.

14. *Which study shows that “human system TASL deficiency fully-phenocopied SLC15A4 loss” (lines146-147)? Please add reference.*

We thank the Reviewer for spotting this oversight. This statement refers to our publications: Heinz *et al*, Nature 2020 (ref. 38) and Zhang *et al*, Cell Reports 2023 (ref. 47).

15. *The experiments in Fig. 3c are based on overexpression of TASL2 in TASL-deficient cells, as such cannot be concluded that TASL2 can substitute TASL, unless TASL2 is expressed at normal levels. Overexpression studies to study the function of TASL2 protein is not the best approach*

since it can lead to interactions that do not happen under normal expression levels and thus wrong conclusions.

We agree with the Reviewer on the general statement that overexpression experiments should be carefully controlled, but we strongly believe that this is the case for the data we obtained in Raw cells presented in Figures 3c, 3d, S3f and S3g when considered altogether. First, to avoid possible variability induced by transient overexpression, we used lentiviral transduction to generate Raw cell lines stably expressing C-tag fused TASL or TASL2 from the same promoter, resulting in comparable protein expression of the two paralogues (Fig. 3c and Fig. S3f). Moreover, in these figures:

- We compared side-by-side TASL-deficient cells stably reconstituted with TASL as well as TASL2 (Fig. 3c), which show that both paralogs have a very strong ability to rescue IRF5 activation when expressed at comparable levels.

- We demonstrated in Figures 3d and S3g that the activity of TASL2 is strictly dependent on the IRF5-binding pLxIS motif and on the first N-terminal amino acids (deltaN construct: deletion of only the amino acids 2-8). The N-terminus of human TASL mediates the binding to SLC15A4 (Heinz *et al.*, Nature 2020), as further confirmed by our recent CryoEM structure data (Chen *et al.*, Nat Commun. 2023, Boeszoermyeni *et al.*, Nat Commun. 2023). Accordingly, we showed in Figures 3a and 3b that this N-terminal region is also crucial for TASL2 binding to mSLC15A4. Given that these three TASL2 constructs are all expressed at comparable levels, we think that the data showed in Fig. 3d strongly support the specificity of the interaction and the conclusion that TASL2 has the potential to substitute TASL and therefore justified further investigation *in vivo* by generating TASL2^{KO} mice. Indeed, the ability of TASL2 to partially compensate for TASL has been strongly supported by the data obtained *ex vivo* and *in vivo* investigating TASL^{KO}, TASL2^{KO} and TASL^{DKO} mice.

16. Clarify the two different wild-type groups (WT and WT control) that appear in Figure 5 and 6. Moreover, statistics are missing in Fig 2f and g.

We agree with the Reviewer that this was not clear. We have now stated in the figure legends that “WT control” group represents non-stimulated/vehicle treated controls (either PBS or PBS+DOTAP injection). Missing statistics in Fig. 2f and g were added.

Reviewer 2:

Drobek et al. provide a detailed analysis of the role of TASL, along with their newly described TASL2 (expressed in mice, not in humans), in immune system signaling and response in TASL, TASL2 and TASL/TASL2 double knock out mice after TLR7 and TLR9 activation. They particularly focus on the role of TASL in IRF5 activation and cytokine and IgG2 production in vitro and in vivo. They also study the role of TASL and TASL2 in vivo after LCMV infection or pristane injection to investigate those components of endosomal TLR signaling pathways in relevant virus infections and in a model of lupus-like disease.

The experiments are nicely performed with data generally compared to data from wild type mice and the "feeble" SLC15A4-deficient mouse. Overall, the data implicate the SLC15A4/TASL complex in IRF5 activation, but not NFkB activation, with comparable results presented for the feeble mouse and the TASL/TASL2 double knockout mouse. Data are more variable for TASL vs. TASL2. Although TASL appears to have a dominant effect compared to TASL2 for most

experimental outcomes, there are differences noted (and not always emphasized in the text) with regard to cell distribution of the two gene products (Fig. S3c and d) and response to in vivo administration of TLR7 (848) vs. TLR9 (CpG) ligands (Figs. 5 and S5). As it appears that there are some differences in immune mediators induced depending on whether signals are provided by TLR7 vs. TLR9 ligands, it would be helpful if the authors could critically describe any differences revealed with regard to those two pathways and the relative contributions of TASL vs. TASL2 in cytokine and IgG induction.

We thank the Reviewer for acknowledging the quality of the data presented and for the constructive comments which helped to further improve our study. Concerning the relative expression and contribution of TASL and TASL2 across cell types and following also the suggestions of Reviewer 1, we included in the revised version additional experiments addressing this point. First, we expanded the direct assessment of the relative expression in splenic B cells and splenic macrophages (modified Fig. S3d). Moreover, we investigated the functional contribution of TASL and TASL 2 in additional cell types, namely conventional DCs (splenic and BM-derived) and splenic macrophages (new Fig. S5a-e). These data further confirmed that TASL^{DKO} phenocopied *feeble*, while single TASL^{KO} and TASL2^{KO} had a partial effect depending on cell type and cytokine assessed (Results lines 227-235). The different impact of TASL and TASL2 in cell type-dependent response and cytokine/IgG production has been further discussed in the revised manuscript (lines 362-375 and 375-388).

As pointed out by the Reviewer, we observed across different experiments a trend suggesting stronger impact of TASL2 deficiency upon CpG-B treatment compared to R848. While we believe that deeper investigation of this potential specific effect on TLR7 vs TLR9 signaling goes beyond the scope of this study, we have now mentioned this point in revised discussion of manuscript as one of the interesting questions to be addressed in follow-up studies (line 373-375).

The two murine systems chosen for study of host defense and autoimmunity, LCMV infection and pristane-induced lupus, are both relevant to TLR7 stimulation. For that reason the models are informative for investigation of TASL. But it is important that the authors strongly make the point that infection with other viruses or other murine lupus models might show very different requirements for TASL and IRF5 in immune activation and tissue pathology.

We thank the Reviewer for supporting the choice and relevance of the two *in vivo* models that we selected. We fully agree with Reviewer's comments on the fact that the role of TASL and IRF5 is likely to depend on the degree by which the investigated viral or autoimmune models are dependent on endosomal TLRs. We amended our discussion accordingly (lines 414-418) to mention more clearly that immune sensing of other viruses which relies more on cytoplasmic nucleic acid sensors (such as RLR/MAVS and cGAS/STING pathways) and less on endosomal TLRs will be most likely less/not affected by deficiency in the SLC15A4/TASL pathway. Similarly, while IRF5 has been shown to be required in a wide range of murine SLE models (Udalova *et al*, Transl Res 2016, PMID: 26207886), we acknowledged that this may not be the case in all SLE-related autoimmune models, such as the one in which the cGAS/STING pathway is central (for example upon TREX1 deficiency). We amended our discussion accordingly (lines 424-430).

With regard to the pristane lupus data shown in Fig. 7, the images for ANA (panel d) and immunoglobulin deposition in the kidney (panel f) are inadequate. Other than the control pristane data the images appear totally black, which does not provide confidence that the cell staining for ANA and tissue sections and staining for the IgG deposition in the kidney are appropriately prepared. In any case, if the authors intend to convincingly show the dramatic abrogation of

autoimmune disease pathology that they suggest, they should also provide data on proteinuria, the typical measure of renal pathology in murine lupus studies. In addition, while the data for anti-Sm/RNP are dramatic, suggesting a strong dependence on TASL and effective TLR7 signaling for production of that specificity (for an RNA-binding protein antigen), it would be helpful to also specifically measure anti-dsDNA autoantibodies in the different experimental conditions. The contribution of TLR7 vs. TLR9 to development of clinical lupus disease based on signaling through each of those endosomal TLR signaling pathways and on development of RNA vs. DNA autoantibody specificities is of great interest in understanding mechanisms of development of autoimmunity in SLE. In some models TLR9 provides protection from autoimmunity while TLR7 drives both type I interferon and autoantibody production. As the authors have established systems to separately use TLR7 and TLR9 stimuli it will be important to elaborate on any distinctions noted in cytokine and IgG production after administering TLR7 vs. TLR9 stimuli (the data in Figs. 5 and S5) and to provide data on RNA-targeted (anti-Sm/RNP) vs. DNA-targeted (anti-dsDNA) autoantibodies in the pristane model experiments.

Regarding the concerns of the Reviewer related to the images illustrating ANA levels and immunoglobulin deposition in the kidneys, we would like to first clarify that the analysis of serum from WT and KO mice were performed at the same time and analysed with the same exact settings. Moreover, for the ANA staining, we paid attention to have samples from different genotypes on each slide (i.e. instead of one genotype per slide) to avoid any possible artifact due to slide-to-slide variability. Therefore, the strong difference between WT and KO groups reflects the strong impairment of autoantibody production and not a technical issue. For the ANA staining, the images of the KOs appear black (i.e no signal) due to the fact that we initially performed a careful titration, with serial dilutions of serum from the pristane-injected WT (which have highest levels of autoantibodies) to avoid oversaturation of the signal (See figure for the Reviewer here below). Similarly, for the IgG deposition in the kidney, the microscopy settings were set to avoid saturation of signal and a noisy background in WT, and then we applied these settings to all samples. We understand that the original images could rise questions, we therefore included in the revised version images acquired with more sensitive settings, which increased the signal and allowed to detect the very low-level present in the KO samples. Moreover, we included additional stainings to visualize the cells or tissue: nuclear staining with DAPI for the ANA detection (new Fig. 7d and S8c) and anti-podocin staining to delineate the glomeruli for the IgG deposition analysis (new Fig. 7f).

Concerning the autoantibodies, we agree with the Reviewer' suggestion, and we have now included the assessment of anti-RNA and anti-DNA autoantibodies, as well as an additional evaluation of anti-Su autoantibodies, targeting cytoplasmic antigen Ago2. Altogether, these data confirmed the strong impairment in pristane-induced autoantibodies production in TASL^{KO} and TASL^{DKO}, which phenocopied the reduction observed in *feeble* mice (new Fig. 7c). Regarding the proteinuria, we would like to mention that it is well described that the C57Bl/6J strain (in which all our mouse lines were generated) is relatively resistant to development of kidney pathology induced by immunocomplex deposition in this pristane-induced model and IgG deposition in the glomeruli does not result in strong kidney pathology, which is observed in other strains (such as BALB/c) (Reeves *et al.* Trends Immunol 2009, PMID: 19699150). In agreement with the literature and with previous data on Slc15a4 KO C57Bl/6 mice (Katewa *et al.*, PLOS One, 2021), we could not detect any significant increase proteinuria even in pristane-challenged WT mice. We have now amended the manuscript to take into account this point as well as related Reviewer 1 comments (Reviewer 1, point 7) (lines 315-327).

Considering the new supportive evidence on the specificity of ANA staining and the immunocomplex deposition results, as well as the extended evaluation of different SLE-associated

autoantibodies, we think that, altogether, the data support a clear role the SLC15A4/TASL pathway in this chemically induced SLE model.

Selected dilution for all samples: 300x

Reviewer 2 – Figure 1: Titration of sera from WT and feeble mice 6 months after pristane challenge.

Reviewer #3:

In this paper, the authors explore the SLC15A4-TASL axis and its effects on TLR7-9 induced IRF5 activation following deficiency or knockout. One of the biggest findings of this study is the characterization of TASL and its paralogue TASL2 (Gm6377), demonstrating their distinct contributions to TLR7-9 signaling pathways. The experimental findings, particularly those regarding TASL and TASL2 knockout mouse models, provide compelling evidence for the critical roles of these molecules in modulating immune responses. The observed phenotypic similarities between TASL-deficient, TASL2-deficient, and SLC15A4-deficient mice underscore the importance of TASL-mediated IRF5 activation in regulating TLR7-9-driven inflammatory cascades. Overall, this research provides ground-breaking contributions to the field of immunology, elucidating the intricate mechanisms underlying TLR7-9 signaling and its implications in anti-viral immune clearance as well as for autoimmune pathogenesis.

We thank the Reviewer for acknowledging the ground-breaking nature of our findings and for the constructive feedback, which helped to further improve our study. We have addressed all the comments in the revised version of the manuscript, as detailed in the point-by-point reply here below.

Comments:

1) Unclear to the reviewer why authors used R848 at a concentration of 10 $\mu\text{g/ml}$ in Fig 3c-d, where elsewhere was used at 100 ng/ml.

We used the concentration of 10 $\mu\text{g/ml}$ R848 in all experiments using the murine macrophage cell line RAW 264.7, while the concentration of 100 ng/ml was used for the experiments with primary cells. Indeed, in initial R848 dose-response experiments using RAW 264.7 cells we observed that these cells responded weakly in terms of IRF5 activation, which was maximised only at 5-10 $\mu\text{g/ml}$ R848, in contrast to primary cells where 100 ng/ml was sufficient. The concentration used for these experiments in RAW 264.7 is comparable to the dose of 5-10 $\mu\text{g/ml}$ we previously used in human cell lines Cal1 and THP1 (Heinz *et al.* Nature 2020). Irrespectively of this high concentration, KO of SLC15A4 or TASL strongly impaired IRF5 activation in RAW 264.7 (Fig.S3e), further highlighting that loss of the complex cannot be overcome even with high R848 doses (as we also confirmed in TASL^{KO} BM-pDCs, Fig S2e).

2) Fig1d is missing isotype controls in both mice. Also, while the feeble mice flow staining was lower with IRF7, a significant shift in IRF7 was still observed post R848 treatment in 1d. The authors need to discuss this point and compare the shifts in the signals between times 0 and 5h. This is not clear especially as the authors argue that IRF7 protein induction was diminished in (Fig. 1c-d). Related to this, in western blot following R848 there was no visible difference shown for IRF7 which need to be clarified by the authors. In figure 1c, the authors do not comment on how/why IRF7 levels are similar between wildtype and feeble mice @ 24h of R848 stimulation.

We agree with the Reviewer that IRF7 induction needed some clarification and we have further investigated this point, both by intracellular staining (ICS) and WB analysis. Concerning the ICS, we now provide confirmation of the specificity of the signal by including isotype control as suggested (new Fig 1c-d). As mentioned above while addressing similar concerns from Reviewer 1 (point 5), we further investigated by WB analysis the modest impairment initially observed by including additional timepoints. Indeed, we could confirm the partial, but reproducible reduction in IRF7 induction after both R848 and CpG-B treatment (Fig. S1a-b). In line with this, the transcriptional profiles confirmed the reduction of IRF7 induction in *feeble* BM-pDC (Fig. 4e and S4f). When considered altogether, we think that these data consistently indicate that SLC15A4 deficiency results in partial reduction of IRF7 induction upon TLR7/9 stimulation, which is more evident when assessing the peak of the response and less at later timepoints. The residual induction is most likely due to, on one side, the extreme sensitivity of IRF7 transcriptional regulation to interferons and, on the other, the fact that in *feeble* cells the IFNAR signaling is reduced but not absent as shown by STAT1 phosphorylation (Fig. 1a-b).

3) Fig1s, information to replicate this figure for the reader in S1a-b are lacking.. how long the B cells were incubated in the presence of Brefeldin A (conc'n is missing), how long after r848/ CPG treatment, was BFA added? Why are only 20% of the cells responding to the R848 WRT IL-6 while 10% were positive TNF. Not to mention the very low MFI (100?) was an isotype control used?.. can you show the raw data?

We thank Reviewer for spotting this omission. The missing information has been now included in the Methods of revised manuscript. Briefly, for intracellular cytokine staining, MACS purified B cells were stimulated with R848 or CpG-B, and then Brefeldin A (final concentration of 3 µg/ml) was added after 1h. Cells were incubated overnight before staining and FACS analysis.

Concerning the IL-6 and TNF staining, we agree with the Reviewer that induction of TNF is very weak in B cells, while IL-6 is more abundantly produced as confirmed by ELISA data (Fig. 1e and Fig. S3l). We have included here below the dot-plot data of representative samples to illustrate the raw data requested, as well as isotype control confirming staining specificity (Reviewer 3 – Figure 1 and 2). We would also like to mention that our data are consistent with previous reports investigating TNF and/or IL-6 production in B cells in SLC15A4-deficient mice (Kobayashi *et al.*, *Immunity* 2014, PMID: 25238095; Katewa *et al.*, *PlosONE* 2021, PMID: 33444326).

Reviewer 3 – Figure 1: Representative dot-plots of TNF and IL-6 intracellular staining in purified splenic B cells stimulated either with R848 or CpG-B o/n in the presence of Brefeldin A. Example from data shown in Fig. S1e-f.

Reviewer 3 – Figure 2: Representative dot-plots of purified splenic B cells stimulated either with R848 or CpG-B o/n in the presence of Brefeldin A and stained intracellularly for TNF and IL-6 (top), with corresponding isotype controls (bottom).

4) In fig 2d-e, Splenic pDC were activated with R848 in the presence of Brefeldin A, but the authors seem to pick and choose what result to show in D it is % and E is MFI. It will be more consistent to show both for both cytokines and discuss why only 5-10 % are responding to the R848 treatment for example.

We would like to first clarify that, while we selected for the main Figures 2d and 2e only one panel to illustrate the data on the IFN α and TNF stainings, additional data were provided in the original Fig. S2f-g. Indeed, for IFN α , we showed a representative dot-plot in Fig. S2f (now Fig. S2g), in addition to the % of positive cells presented in Fig. 2d. For the TNF induction after 3 hours of stimulation, the original Fig. S2g (now Fig. S2h) showed % of positive cells and a representative histogram to complement the gMFI data shown in Fig2e. Moreover, Fig. S2h (now Fig. S2i) showed both % and gMFI for TNF upon overnight stimulation.

Concerning the 5-10 % of IFN α + cells, we would like to mention that this is in line with previous reports using the same settings (Dosenovic *et al.*, Immunol Cell Biol. 2015, PMID: 25310967; Abbas *et al.*, Nat Immunol 2020, PMID: 32690951). Indeed, only in a small portion of pDCs is IFN α usually detectable by intracellular FACS staining. For this reason, we have included the representative dot plot (instead of the gMFI) as supplementary figure (former Fig. S2f, now Fig.S2g). Lastly, the impairment in IFN α observed by ICS in TASL^{KO} is fully consistent with results obtained by qPCR data (Fig. 2f-g).

5) LCMV cytotoxic T cell responses is key in LCMV clearance, yet the authors could not detect differences as measured by LCMV-specific tetramers which suggests other functional assays would provide better answers. This should be discussed.

We thank the Reviewer for the suggestion to assess CD8+ T cell function in addition to the measurement of LCMV-specific tetramers, which we have now included in the revised manuscript (new Fig. 6d-e and Fig. S7e-g). To address functional response of cytotoxic CD8 T cells, we focused on *in vitro* restimulation with LCMV immunodominant peptides as it was previously reported that the main functional impairment observed in TLR7 deficient mice upon chronic LCMV infection was a strong reduction in double (IFN γ , TNF) and triple cytokine producers (IFN γ , TNF, IL-2) (Walsh et al, Cell Host Microbe 2012, PMID: 22704624). Indeed, in TASL^{DKO} and *feeble* mice we observed strongly reduced frequency of these triple-producers resulting in very few cytokine-secreting CD8+ T cells in absolute numbers, with single TASL^{KO} and TASL2^{KO} showing still a significant impairment, but intermediate phenotypes (new Fig. 6e and Fig. S7g). This defect was detectable with reduction in double-producers early after infection (day 8, Fig. 6d and Fig.S7e) as well as double- and triple-producers in later timepoints (90 days, Fig. 6e and Fig S7f-g). Together with reduced B cell responses and lower LCMV-specific antibody titers in blood, we think that this defect can further contribute to explain the observed viral persistence in TASL^{DKO} and *feeble* mice. We thank the Reviewer for the important suggestion that helped to improve our study.

REVIEWER COMMENTS

Reviewer #1 (Remarks to the Author):

In the revised version of the manuscript the authors addressed many of my comments, but still show a strong bias on emphasizing the role of TASL, while trying to minimize the role of TASL2, both while presenting the data and during the discussion. I agree with the authors that TASL/TASL2 double deficiency phenocopies SLC15A4 deficiency, but it is also important to clearly present in an unbiased manner the independent contribution of TASL and TASL2 in the various assays and cell types. The authors are the first ones that identified TASL2 and generated deficient mice and so they are in the great position to clarify the independent function of TASL and TASL2. Moreover, based on the fact that in the various studies both sexes of mice were mixed is raising an important issue regarding the validity of the data.

We thank the reviewer for the assessment of revised version of our manuscript and we are pleased that she/he acknowledges that we addressed many of the comments.

*We have detailed in the point-by-point reply here below the answers to the remaining concerns and the changes that we included in the new revised version of the manuscript. Concerning the role of TASL and TASL2, we have now extended the description of the relative contribution of the two paralogues observed in our experiments (as requested in point 1) to be more precise on the effect of TASL2^{KO}. We would like to mention that it was not our intention to minimize the role of TASL2: our comments on the stronger effect of TASL deficiency compared to TASL2 deficiency reflected the overall effects observed in our study, including the “black and white” results we obtained in the *in vivo* experiments upon injection with TLR ligands or pristane. Indeed, as shown in Fig. 5 and S6, TASL^{KO} phenocopy TASL^{DKO} displaying a profound impairment in inflammatory responses upon both R848 or CpG injections, while TASL2^{KO} behave largely as WT. Similarly, inflammatory monocyte infiltration and the ISG signature upon pristane peritoneal injection is blunted in TASL^{KO} (as in TASL^{DKO}), but largely unaffected in TASL2^{KO} (Fig. 7a-b and S8b). Concerning the sexes of the mice used, we believe this is misunderstanding due to our unclear description in the Figure legends, which we have now corrected. As described below, we now provide in each figure legend the detailed description of the sex of the animals used for each experiment, which address this concern and clearly demonstrate the validity of our data.*

Points that still have to be addressed:

1. The new data that were generated by answering my previous Points 1-4 regarding the differential redundancy between TASL and TASL2 in the various cell types including cDCs, macrophages and B cells should be appropriately presented and discussed in the manuscript and not only in the point-by-point response to the reviewer.

As requested, in the revised manuscript we extended the description of these data and further verified that this reflected the presentation given in the detailed point-by-point response.

2. Previous points 6 and 7 have not been fully addressed. As the authors mention pristane-lupus model is not ideal in the C57BL/6 background, thus at least maximum possible data are needed to support correct conclusions. In their rebuttal letter the authors responded “We would like to point out that the conclusion that TASLKO has more profound impact than TASL2KO is also very

strongly supported by the in vivo Results”, however, no in vivo data are provided for the TSL2ko mice for certain experiments. The contribution of TSL2 in lupus development should be clarified by presenting sera autoantibodies, ANA antibodies and kidney histology from TSL2ko mice, as has been requested in the revision of the original manuscript and as it has been done for TSL deficient mice (Figure 7 c-g). Moreover, histopathological scoring for glomerulonephritis in the kidneys of all mouse groups has to be done by a pathologist. The data provided in Figure 3 in the rebuttal letter are not convincing, magnification is too low and scoring not detailed. The argument of the authors that previous studies did not reveal glomerulonephritis in SLC15A4 deficient mice is not valid since microbiota plays an important role in autoimmune development and the microbiota vary dramatically from one animal facility to another.

We would like to first clarify that we strongly believe that the conclusions regarding the pristane model that we made in our manuscript are fully supported by the data. Indeed, these are:

- Single TSL deficiency is sufficient to phenocopy feeble as well as TSL^{DKO}, strongly impairing all the inflammatory and autoimmune manifestations that can be assessed in this model (both in the short and long-term experiments).*
- These results demonstrate that TSL2 cannot compensate for TSL loss in this model (in contrast to the LCMV model).*

Concerning the quoted text from our rebuttal letter, we would like to point out that this comment was in the answer to another question (point 1) and primarily related to the in vivo stimulation with TLR7/9 agonists. Indeed, the original full statement reads: "We would like to point out that the conclusion that TSL^{KO} has more profound impact than TSL2^{KO} is also very strongly supported by the in vivo results. Indeed, TSL deficiency had much stronger impact than TSL2^{KO} on all responses assessed upon in vivo TLR7 and TLR9 stimulation (Fig. 5 and S6)." Nevertheless, this is also valid for the short term pristane data (Fig. 7a-b) in which single TSL deficiency blunted recruitment of inflammatory monocytes and ISG signature, while this was not the case for TSL2^{KO}, which showed unaltered responses similar to WT. The normal induction of inflammatory responses in TSL2^{KO} in the short-term experiment (2 weeks) is the reason why we did not investigate the effect of TSL2 deficiency in the long-term experiment (6 months), as we mentioned in previous reply to this point. Importantly, in our manuscript we did not state that TSL2 is necessarily dispensable for the long-term autoimmune manifestation, but focused on the fact that TSL deficiency is sufficient to rescue autoimmune phenotype. This addressed the primary objective of this experiment, which is to demonstrate that the SLC15A4-TSL complex is essential in this chemically induced SLE model and that the protective effect of SLC15A4-deficiency in this model is not mediated by other TSL-independent mechanisms. In the revised manuscript we further clarified this point by explicitly stating that even though no phenotype was observed for TSL2^{KO} in the short-term experiments, this does not imply that TSL2 is necessary dispensable in the long-term (see lines 412-417). For these reasons, we strongly believe that these long-term data are not essential to support our conclusion. Moreover, as mentioned in the previous reply, their inclusion will result in several months of delay (beyond the 3 months allowed for the revision), potentially affecting the timely publication of these novel findings, which demonstrate for the first time the in vivo role of TSL and TSL2.

Concerning the histopathology assessment, we would like to clarify that the scoring was indeed done by pathologists, new authors Dr. Rotman and Dr. Royer-Chardon from the Department of Pathology, Lausanne University Hospital (CHUV). Given that it is widely reported in the literature that the pristane model in C57Bl/6 does not result in significant kidney pathology, we decided to monitor a sensitive readout which is the immune infiltration. This is based on the assessment of inflammatory infiltrate, represented predominantly by mononuclear cells, in the renal cortex (Inflammation score (% of with inflammatory infiltrate): 0 = <10%, 1 = 10-50%; 2= >50%). The evaluation was performed by the pathologists on the full kidney section, and the magnification of

the representative images that we included in the figure 3 for the reviewer was selected to display sufficiently large area to be representative, as the distribution of infiltrating immune cells is not completely homogenous. Blind assessment of this failed to detect any clear increase even in the pristane injected WT mice, which is not surprising, but prevented further assessment of the knockouts. Of note, since conception of this experiment, we planned to assess kidney sections by monitoring immunoglobulin deposition by immunofluorescence (Fig. 7f-g), which is widely used and reported to be robust readout in this model and indeed reveal clear signal in WT and impairment in KOs. This require OCT embedded cryosections which is also compatible with evaluation of immune infiltration but could lead to artefacts when evaluating glomeruli morphology by H&E staining and therefore data of poor quality.

3. Previous Point 11 has not been addressed. Human and mouse genes and proteins are not the same and since the studies performed in mice this should be mentioned in the title of the paper.

To conform with reviewer' request, we have modified the title as follows:

"TLR7/9 murine adaptors TASL and TASL2 mediate IRF5-dependent antiviral responses and autoimmunity."

4. Previous point 12 regarding sex and age of mice used in the studies and the request that this information have to be added in all the Figure legends. In the revised versions and in many figure legends now it is written: "Both males and females were used as a source of primary cells". Mixing genotypes for the various assays is not appropriate since it is not obvious at what degree the differences that are observed between the genotypes are the outcome of the genotype and/or the sex, since immune responses vary depending on the sex. More specific, it is known that pDCs from female mice and humans have higher basal levels of IRF5 and IFN-alpha production following TLR7 stimulation (Griesbeck et al., J Immunol, 2015, 195:5327). Thus, using only females or more females in one genotype than in the other can give a bias for higher TLR7 signaling for this genotype. Moreover, even for the in vivo studies - LCMV infection and pristane -induced lupus - the authors used both sexes. Mixing both sexes it is also not appropriate for the current study since TLR7 is located in the X chromosome both in mice and humans. It is well established that the TLR7 gene, encoded in the X chromosome both in mice and humans, can escape X inactivation resulting is higher expression of TLR7 in females than males that can partially explain the higher incidence of lupus in females (Pisitkun et al., Science, 2006, 312:1669). Thus, the mice used for each experiment have to be of the same sex for all the genotypes.

We acknowledge that the formulation we used in the figure legend ("Both males and females were used as a source of primary cells") was very unfortunate and could lead to confusion. We have now clarified in revised manuscript that experiments were performed with sex-matched cells for each genotype, and have included in the figure legend the numbers and sex of experimental animals used in each panel as requested. Moreover, the information on the sex is now highlighted in the revised source data file for each single datapoint. Of note, the difference reported in WT mice between males and females are quantitatively modest while the differences we observe between our KOs and WT are much more profound ("black and white") in both, males and females (see data for Figure 2c here below). Therefore, the use of mixed groups in a subset of panels has no impact on the results and do not affect the conclusion. Indeed, for the LCMV and pristane experiments the results using only 1 sex reflect the data obtained from combining both sexes (see data for Figures 6 and 7 here below).

Figure 2c - in vitro BM-pDC stimulation for IFN and cytokine production

Manuscript Fig. 2c - Males and Females

Fig. 2c - WT mice only - Males vs Females (effect of sex)

Fig. 2c - WT vs TASL^{KO} (Males)

Fig. 2c - WT vs TASL^{KO} (Females)

Figure 6 - LCMV infection

Manuscript Fig. 6a - Cytokine response

Fig. 6a - One sex only (Females)

Manuscript Fig. 6b - Viral titer

Fig. 6b - One sex only (Males)

Manuscript Fig. 6d - *in vitro* peptide stimulation

Fig. 6d - One sex only (Females)

Manuscript Fig. 6f - B cell response

Fig. 6f - One sex only (Females)

Manuscript Fig. 6g - T_m response

Fig. 6g - One sex only (Females)

Figure 7 - Pristine injection

Manuscript Fig. 7a - Peritoneal infiltration

Fig. 7a - One sex only (Males)

Manuscript Fig. 7b - ISG signature in peritoneal cells

Fig. 7b - One sex only (Males)

Manuscript Fig. S8a - Peritoneal infiltration

Fig. S8a - One sex only (Males)

Reviewer #2 (Remarks to the Author):

The authors have done an excellent job of responding to all reviewer comments. They have performed extensive additional experiments and have very thoughtfully discussed the comments and questions raised by the reviewers. It will be important for future studies to investigate TASL and TASL2 in additional murine models of lupus as well as additional models of virus infection.

We thank the reviewer for the valuable and constructive suggestions which helped to improve our study.

Response to Reviewer 1 (NCOMMS-24-00959B)

Reviewer #1 (Remarks to the Author):

Data on old TLSL2ko mice regarding lupus development (sera autoantibodies, ANA antibodies and kidney histology) still have not been provided. The reply of the authors very clearly explains why, but still does not resolve the issues that I raised in my previous reports.

Moreover, the issue on performing experiments by mixing cells of mice from both sexes remains. Many experiments were performed by mixing female and male cells/mice despite the fact that TLR7 is located on the X chromosome. It is known that female cells show higher response than males upon TLR7 stimulation and in certain degree the increased TLR7 signaling in females is responsible for the higher incidence of lupus in females than males. Based on the argumentation of the authors in their rebuttal letter, I agree that for a phenotype that is profound like in the case of TASLDKO and feeble versus wild type mice mixing both sexes maybe is not so vital. However, for a modest phenotype like in the case of TLSL2ko mice this can have detrimental consequence on driving correct conclusions.

We thank the reviewer for considering our argumentation on these two last points. We would like to restate that the experiments were performed with largely sex matched cells/mice groups, as detailed in the figure legends for each panel as requested, and that any minor difference does not account for the phenotypes observed nor affects the conclusions (as highlighted in the Source data indicating the sex for each single data point). Moreover, the results obtained in the *in vivo* LCMV and pristane models by combining both sexes are virtually identical to the one obtained when considering only one sex, as shown in the side-by-side data figures included in the previous reply.